# Monte Carlo guided Denoising Diffusion Models for Bayesian linear inverse problems

**Gabriel Cardoso***
Ecole polytechnique
IHU Liryc

**Yazid Janati***
Ecole polytechnique

**Sylvain Le Corff**
Sorbonne Université

**Eric Moulines**
Ecole polytechnique

## Abstract

Ill-posed linear inverse problems arise frequently in various applications, from computational photography to medical imaging. A recent line of research exploits Bayesian inference with informative priors to handle the ill-posedness of such problems. Amongst such priors, score-based generative models (SGM) have recently been successfully applied to several different inverse problems. In this paper, we exploit the particular structure of the prior defined by the SGM to define a sequence of intermediate linear inverse problems. As the noise level decreases, the posterior distributions of these inverse problems get closer to the target posterior of the original inverse problem. To sample from these distributions, we propose the use of Sequential Monte Carlo (SMC) methods. The proposed algorithm, MCGdiff, is shown to be theoretically grounded and we provide numerical simulations showing that it outperforms competing baselines when dealing with ill-posed inverse problems in a Bayesian setting.

## 1 Introduction

This paper is concerned with linear inverse problems $y = \mathrm{A}x + \sigma_y \varepsilon$, where $y \in \mathbb{R}^{\mathsf{d}_y}$ is a vector of indirect observations, $x \in \mathbb{R}^{\mathsf{d}_x}$ is the vector of unknowns, $\mathrm{A} \in \mathbb{R}^{\mathsf{d}_y \times \mathsf{d}_x}$ is the linear forward operator and $\varepsilon \in \mathbb{R}^{\mathsf{d}_y}$ is an unknown noise vector. This general model is used throughout computational imaging, including various tomographic imaging applications such as common types of magnetic resonance imaging Vlaardingerbroek & Boer (2013), X-ray computed tomography Elbakri & Fessler (2002), radar imaging Cheney & Borden (2009), and basic image restoration tasks such as deblurring, super-resolution, and image inpainting González et al. (2009). The classical approach to solving linear inverse problems relies on prior knowledge about $x$, such as its smoothness, sparseness in a dictionary, or its geometric properties. These approaches attempt to estimate a $\hat{x}$ by minimizing a regularized inverse problem, $\hat{x} = \mathrm{argmin}_x \{\|y - \mathrm{A}x\|^2 + \mathrm{Reg}(x)\}$, where $\mathrm{Reg}$ is a regularization term that balances data fidelity and noise while enabling efficient computations. However, a common difficulty in the regularized inverse problem is the selection of an appropriate regularizer, which has a decisive influence on the quality of the reconstruction.

Whereas regularized inverse problems continue to dominate the field, many alternative statistical formulations have been proposed; see Besag et al. (1991); Idier (2013); Marnissi et al. (2017) and the references therein - see Stuart (2010) for a mathematical perspective. A main advantage of statistical approaches is that they allow for **uncertainty quantification** in the reconstructed solution; see Dashti & Stuart (2017). The **Bayes' formulation** of the regularized inverse problem is based on considering the indirect measurement $Y$, the state $X$ and the noise $\varepsilon$ as random variables, and to specify $p(y|x)$ the *likelihood* (the conditional distribution of $Y$ at $X$) and the prior $p(x)$ (the distribution of the state). One can use Bayes' theorem to obtain the **posterior distribution** $p(x|y) \propto p(y|x)p(x)$, where "$\propto$" means that the two sides are equal to each other up to a multiplicative constant that does not depend on $x$. Moreover, the use of an appropriate method for Bayesian inference allows the

---

*Both authors contributed equally.
Correspondence: {gabriel.victorino_cardoso,yazid.janati}@polytechnique.edu

quantification of the uncertainty in the reconstructed solution $x$. A variety of priors are available, including but not limited to Laplace Figueiredo et al. (2007), total variation (TV) Kaipio et al. (2000) and mixture-of-Gaussians Fergus et al. (2006). In the last decade, a variety of techniques have been proposed to design and train generative models capable of producing perceptually realistic samples from the original data, even in challenging high-dimensional data such as images or audio Kingma et al. (2019); Kobyzev et al. (2020); Gui et al. (2021). Denoising diffusion models have been shown to be particularly effective generative models in this context Sohl-Dickstein et al. (2015); Song et al. (2021c;a;b); Benton et al. (2022). These models convert noise into the original data domain through a series of denoising steps. A popular approach is to use a generic diffusion model that has been pre-trained, eliminating the need for re-training and making the process more efficient and versatile Trippe et al. (2023); Zhang et al. (2023). Although this was not the main motivation for developing these models, they can of course be used as prior distributions in Bayesian inverse problems. This simple observation has led to a new, fast-growing line of research on how linear inverse problems can benefit from the flexibility and expressive power of the recently introduced deep generative models; see Arjomand Bigdeli et al. (2017); Wei et al. (2022); Su et al. (2022); Kaltenbach et al. (2023); Shin & Choi (2023); Zhihang et al. (2023); Sahlström & Tarvainen (2023).

## CONTRIBUTIONS

• We propose `MCGdiff`, a novel algorithm for sampling from the Bayesian posterior of Gaussian linear inverse problems with denoising diffusion model priors. `MCGdiff` specifically exploits the structure of both the linear inverse problem and the denoising diffusion generative model to design an efficient SMC sampler.

• We establish under sensible assumptions that the empirical distribution of the samples produced by `MCGdiff` converges to the target posterior when the number of particles goes to infinity. To the best of our knowledge, `MCGdiff` is the first provably consistent algorithm for conditional sampling from the denoising diffusion posteriors.

• To evaluate the performance of `MCGdiff`, we perform numerical simulations on several examples for which the target posterior distribution is known. Simulation results support our theoretical results, i.e. the empirical distribution of samples from `MCGdiff` converges to the target posterior distribution. This is **not** the case for the competing methods (using the same denoising diffusion generative priors) which are shown, when run with random initialization of the denoising diffusion, to generate a significant number of samples outside the support of the target posterior. We also illustrate samples from `MCGdiff` in imaging inverse problems.

**Background and notations.** This section provides a concise overview of the diffusion model framework and notations used in this paper. We cover the elements that are important for understanding our approach, and we recommend that readers refer to the original papers for complete details and derivations Sohl-Dickstein et al. (2015); Ho et al. (2020); Song et al. (2021c;a). A denoising diffusion model is a generative model consisting of a forward and a backward process. The forward process involves sampling $X_0 \sim q_{\text{data}}$ from the data distribution, which is then converted to a sequence $X_{1:n}$ of recursively corrupted versions of $X_0$. The backward process involves sampling $X_n$ according to an easy-to-sample reference distribution on $\mathbb{R}^{d_x}$ and generating $X_0 \in \mathbb{R}^{d_x}$ by a sequence of denoising steps. Following Sohl-Dickstein et al. (2015); Song et al. (2021a), the forward process can be chosen as a Markov chain with joint distribution

$$q_{0:n}(x_{0:n}) = q_{\text{data}}(x_0) \prod_{t=1}^{n} q_t(x_t|x_{t-1}), \quad q_t(x_t|x_{t-1}) = \mathcal{N}(x_t; (1-\beta_t)^{1/2} x_{t-1}, \beta_t I_{d_x}), \quad (1.1)$$

where $I_{d_x}$ is the identity matrix of size $d_x$, $\{\beta_t\}_{t\in\mathbb{N}} \subset (0,1)$ is a non-increasing sequence and $\mathcal{N}(\mathbf{x}; \mu, \Sigma)$ is the p.d.f. of the Gaussian distribution with mean $\mu$ and covariance matrix $\Sigma$ (assumed to be non-singular) evaluated at $\mathbf{x}$. For all $t > 0$, set $\bar{\alpha}_t = \prod_{\ell=1}^{t}(1-\beta_\ell)$ with the convention $\alpha_0 = 1$. We have for all $0 \le s < t \le n$,

$$q_{t|s}(x_t|x_s) := \int \prod_{\ell=s+1}^{t} q_\ell(x_\ell|x_{\ell-1}) dx_{s+1:t-1} = \mathcal{N}(x_t; (\bar{\alpha}_t/\bar{\alpha}_s)^{1/2} x_s, (1-\bar{\alpha}_t/\bar{\alpha}_s) I_{d_x}). \quad (1.2)$$

For the standard choices of $\bar{\alpha}_t$, the sequence of distributions $(q_t)_{t\in\mathbb{N}}$ converges weakly to the standard normal distribution as $t \to \infty$, which we chose as the reference distribution. For the reverse process, Song et al. (2021a;b) introduce an *inference distribution* $q^{\sigma}_{1:n|0}(x_{1:n}|x_0)$, depending on a sequence $\{\sigma_t\}_{t\in\mathbb{N}}$ of hyperparameters satisfy-

ing $\sigma_t^2 \in [0, 1 - \bar{\alpha}_{t-1}]$ for all $t \in \mathbb{N}^*$, and defined as $q_{1:n|0}^\sigma(x_{1:n}|x_0) = q_{n|0}^\sigma(x_n|x_0) \prod_{t=n}^2 q_{t-1|t,0}^\sigma(x_{t-1}|x_t, x_0)$, where $q_{n|0}^\sigma(x_n|x_0) = \mathcal{N}\left(x_n; \bar{\alpha}_n^{1/2} x_0, (1 - \bar{\alpha}_n) \mathrm{I}_{\mathsf{d}_x}\right)$ and $q_{t-1|t,0}^\sigma(x_{t-1}|x_t, x_0) = \mathcal{N}\left(x_{t-1}; \boldsymbol{\mu}_t(x_0, x_t), \sigma_t^2 \mathrm{I}_{\mathsf{d}_x}\right)$, with $\boldsymbol{\mu}_t(x_0, x_t) = \bar{\alpha}_{t-1}^{1/2} x_0 + (1 - \bar{\alpha}_{t-1} - \sigma_t^2)^{1/2}(x_t - \bar{\alpha}_t^{1/2} x_0)/(1 - \bar{\alpha}_t)^{1/2}$. For $t \in [1 : n-1]$, we define by backward induction the sequence $q_{t|0}^\sigma(x_t|x_0) = \int q_{t|t+1,0}^\sigma(x_t|x_{t+1}, x_0) q_{t+1|0}^\sigma(x_{t+1}|x_0) \mathrm{d}x_{t+1}$. It is shown in (Song et al., 2021a, Lemma 1) that for all $t \in [1 : n]$, the distributions of the forward and inference process conditioned on the initial state coincide, i.e. that $q_{t|0}^\sigma(x_t|x_0) = q_{t|0}(x_t|x_0)$. The backward process is derived from the inference distribution by replacing, for each $t \in [2 : n]$, $x_0$ in the definition $q_{t-1|t,0}^\sigma(x_{t-1}|x_t, x_0)$ with a prediction where $\boldsymbol{\chi}_{0|t}^\theta(x_t) := \bar{\alpha}_t^{-1/2}\left(x_t - (1 - \bar{\alpha}_t)^{1/2} \mathbf{e}^\theta(x_t, t)\right)$ where $\mathbf{e}^\theta(x, t)$ is typically a neural network parameterized by $\theta$. More formally, the backward distribution is defined as $\mathsf{p}_{0:n}^\theta(x_{0:n}) = \mathsf{p}_n(x_n) \prod_{t=0}^{n-1} p_t^\theta(x_t|x_{t+1})$, where $\mathsf{p}_n(x_n) = \mathcal{N}(x_n; 0_{\mathsf{d}_x}, \mathrm{I}_{\mathsf{d}_x})$ and for all $t \in [1 : n-1]$,

$$p_t^\theta(x_t|x_{t+1}) := q_{t|t+1,0}^\sigma(x_t|x_{t+1}, \boldsymbol{\chi}_{0|t+1}^\theta(x_{t+1})) = \mathcal{N}(x_t, \mathsf{m}_{t+1}^\theta(x_{t+1}), \sigma_{t+1}^2 \mathrm{I}_{\mathsf{d}_x}), \tag{1.3}$$

where $\mathsf{m}_{t+1}(x_{t+1}) := \boldsymbol{\mu}(\boldsymbol{\chi}_{0|t+1}^\theta(x_{t+1}), x_{t+1})$ and $0_{\mathsf{d}_x}$ is the null vector of size $\mathsf{d}_x$. At step 0, we set $p_0(x_0|x_1) := \mathcal{N}(x_0; \boldsymbol{\chi}_{0|1}^\theta(x_1), \sigma_1^2 \mathrm{I}_{\mathsf{d}_x})$. The parameter $\theta$ is obtained (Song et al., 2021a, Theorem 1) by solving the following optimization problem:

$$\theta_* \in \arg\min_\theta \sum_{t=1}^n (2\mathsf{d}_x \sigma_t^2 \alpha_t)^{-1} \int \|\epsilon - \mathbf{e}^\theta(\sqrt{\alpha_t} x_0 + \sqrt{1 - \alpha_t}\epsilon, t)\|_2^2 \mathcal{N}(\epsilon; 0_{\mathsf{d}_x}, \mathrm{I}_{\mathsf{d}_x}) \mathsf{q}_{\mathrm{data}}(\mathrm{d}x_0) \mathrm{d}\epsilon. \tag{1.4}$$

Thus, $\mathbf{e}^{\theta_*}(X_t, t)$ might be seen as the predictor of the noise added to $X_0$ to obtain $X_t$ (in the forward pass) and justifies the "prediction" terminology. The time 0 marginal $\mathsf{p}_0^{\theta_*}(x_0) = \int \mathsf{p}_{0:n}^{\theta_*}(x_{0:n}) \mathrm{d}x_{1:n}$ which we will refer to as the *prior* is used as an approximation of $\mathsf{q}_{\mathrm{data}}$ and the time $s$ marginal is $\mathsf{p}_s^{\theta_*}(x_s) = \int \mathsf{p}_{0:n}^{\theta_*}(x_{0:n}) \mathrm{d}x_{1:s-1} \mathrm{d}x_{s+1:n}$. In the rest of the paper, we drop the dependence on the parameter $\theta_*$. We define for all $v \in \mathbb{R}^\ell, w \in \mathbb{R}^k$, the concatenation operator $v^\frown w = [v^T, w^T]^T \in \mathbb{R}^{\ell+k}$. For $i \in [1 : \ell]$, we let $v[i]$ the $i$-th coordinate of $v$.

**Related works.** The subject of Bayesian problems is very vast, and it is impossible to discuss here all the results obtained in this very rich literature. One of such domains is image restoration problems, such as deblurring, denoising inpainting, which are challenging problems in computer vision that involves restoring a partially observed degraded image. Deep learning techniques are widely used for this task Arjomand Bigdeli et al. (2017); Yeh et al. (2018); Xiang et al. (2023); Wei et al. (2022) with many of them relying on auto-encoders, VAEs Ivanov et al. (2018); Peng et al. (2021); Zheng et al. (2019), GANs Yeh et al. (2018); Zeng et al. (2022), or autoregressive transformers Yu et al. (2018); Wan et al. (2021). In what follows, we focus on methods based on denoising diffusion that has recently emerged as a way to produce high-quality realistic samples from the original data distribution on par with the best GANs in terms of image and audio generation, without the intricacies of adversarial training; see Sohl-Dickstein et al. (2015); Song et al. (2021c; 2022). Diffusion-based approaches do not require specific training for degradation types, making them much more versatile and computationally efficient. In Song et al. (2022), noisy linear inverse problems are proposed to be solved by diffusing the degraded observation forward, leading to intermediate observations $\{y_s\}_{s=0}^n$, and then running a modified backward process that promotes consistency with $y_s$ at each step $s$. The Denoising-Diffusion-Restoration model (DDRM) Kawar et al. also modifies the backward process so that the unobserved part of the state follows the backward process while the observed part is obtained as a noisy weighted sum between the noisy observation and the prediction of the state. As observed by Lugmayr et al. (2022), DDRM is very efficient, but the simple blending used occasionally causes inconsistency in the restoration process. DPS Chung et al. (2023) considers a backward process targeting the posterior. DPS approximates the score of the posterior using the Tweedie formula, which incorporates the learned score of the prior. The approximation error is quantified and shown to decrease when the noise level is large, i.e., when the posterior is close to the prior distribution. As shown in Section 3 with a very simple example, neither DDRM nor DPS can be used to sample the target posterior and therefore do not solve the Bayesian recovery problem (even if we run DDRM and DPS several time with independent initializations). Indeed, we show that DDRM and DPS produce samples under the "prior" distribution (which is generally captured very well by the denoising diffusion model), but which are not consistent with the observations (many samples land in areas with very low likelihood).

In Trippe et al. (2023), the authors introduce SMCdiff, a Sequential Monte Carlo-based denoising diffusion model that aims at solving specifically the *inpainting problem*. SMCdiff produces a particle approximation of the conditional distribution of the non observed part of the state conditionally on a forward-diffused trajectory of the observation. The resulting particle approximation is shown to converge to the true posterior of the SGM under the assumption that the joint laws of the forward and backward processes coincide, which fails to be true in realistic setting. In comparison with SMCdiff, MCGdiff is a versatile approach that solves any Bayesian linear inverse problem while being consistent under mild assumptions. In parallel to our work, Wu et al. (2023) also developed a similar SMC based methodology but with a different proposal kernel.

## 2   THE MCGDIFF ALGORITHM

In this section, we present our methodology for the inpainting problem equation 2.1, both with noise and without noise. The more general case is treated in Section 2.1. Let $\mathsf{d}_y \in [1 : \mathsf{d}_x - 1]$. In what follows we denote the $\mathsf{d}_y$ top coordinates of a vector $x \in \mathbb{R}^{\mathsf{d}_x}$ by $\overline{x}$ and the remaining coordinates by $\underline{x}$, so that $x = \overline{x}^\frown\underline{x}$. The inpainting problem is defined as

$$Y = \overline{X} + \sigma_y\varepsilon, \quad \varepsilon \sim \mathcal{N}(0, \mathrm{I}_{\mathsf{d}_y}), \quad \sigma \geq 0, \tag{2.1}$$

where $\overline{X}$ are the first $\mathsf{d}_y$ coordinates of a random variable $X \sim \mathsf{p}_0$. The goal is then to recover the law of the complete state $X$ given a realisation $\mathbf{y}$ of the incomplete observation $\mathbf{Y}$ and the model equation 2.1.

**Noiseless case.** We begin by the case $\sigma_y = 0$. As the first $\mathsf{d}_y$ coordinates are observed exactly, we aim at infering the remaining coordinates of $X$, which correspond to $\underline{X}$. As such, given an observation $y$, we aim at sampling from the posterior $\phi_0^y(\underline{x}_0) \propto \mathsf{p}_0(y^\frown\underline{x}_0)$ with integral form

$$\phi_0^y(\underline{x}_0) \propto \int \mathsf{p}_n(x_n) \left\{ \prod_{s=1}^{n-1} p_s(x_s|x_{s+1}) \right\} p_0(y^\frown\underline{x}_0|x_1)\mathrm{d}x_{1:n}. \tag{2.2}$$

To solve this problem, we propose to use SMC algorithms Doucet et al. (2001); Cappé et al. (2005); Chopin et al. (2020), where a set of $N$ random samples, referred to as particles, is iteratively updated to approximate the posterior distribution. The updates involve, at iteration $s$, selecting promising particles from the pool of particles $\xi_{s+1}^{1:N} = (\xi_{s+1}^1, \ldots, \xi_{s+1}^N)$ based on a weight function $\widetilde{\omega}_s$, and then apply a Markov transition $p_s^y$ to obtain the samples $\xi_s^{1:N}$. The transition $p_s^y(x_s|x_{s+1})$ is designed to follow the backward process while guiding the $\mathsf{d}_y$ top coordinates of the pool of particles $\xi_s^{1:N}$ towards the measurement $y$. Note that under the backward dynamics equation 1.3, $\overline{X}_t$ and $\underline{X}_t$ are independent conditionally on $X_{t+1}$ with transition kernels respectively $\overline{p}_t(\overline{x}_t|x_{t+1}) := \mathcal{N}(\overline{x}_t; \overline{\mathsf{m}}_{t+1}(x_{t+1}), \sigma_{t+1}^2\mathrm{I}_{\mathsf{d}_y})$ and $\underline{p}_t(\underline{x}_t|x_{t+1}) := \mathcal{N}(\underline{x}_t; \underline{\mathsf{m}}_{t+1}(x_{t+1}), \sigma_{t+1}^2\mathrm{I}_{\mathsf{d}_x-\mathsf{d}_y})$ where $\overline{\mathsf{m}}_{t+1}(x_{t+1}) \in \mathbb{R}^{\mathsf{d}_y}$ and $\underline{\mathsf{m}}_{t+1}(x_{t+1}) \in \mathbb{R}^{\mathsf{d}_x-\mathsf{d}_y}$ are such that $\mathsf{m}_{t+1}(x_{t+1}) = \overline{\mathsf{m}}_{t+1}(x_{t+1})^\frown\underline{\mathsf{m}}_{t+1}(x_{t+1})$ and the above kernels satisfy $p_t(x_t|x_{t+1}) = \overline{p}_t(\overline{x}_t|x_{t+1})\underline{p}_t(\underline{x}_t|x_{t+1})$. We consider the following proposal kernels for $t \in [1 : n - 1]$,

$$p_t^y(x_t|x_{t+1}) \propto p_t(x_t|x_{t+1})\overline{q}_{t|0}(\overline{x}_t|y), \quad \text{where} \quad \overline{q}_{t|0}(\overline{x}_t|y) := \mathcal{N}(\overline{x}_t; \bar{\alpha}_t^{1/2}y, (1 - \bar{\alpha}_t)\mathrm{I}_{\mathsf{d}_y}). \tag{2.3}$$

For the final step, we define $\underline{p}_0^y(\underline{x}_0|x_1) = \underline{p}_0(\underline{x}_0|x_1)$. Using standard Gaussian conjugation formulas, we obtain

$$p_t^y(x_t|x_{t+1}) = \underline{p}_t(\underline{x}_t|x_{t+1}) \cdot \mathcal{N}\left(\overline{x}_t; \mathsf{K}_t\alpha_t^{1/2}y + (1 - \mathsf{K}_t)\overline{\mathsf{m}}_{t+1}(x_{t+1}), (1 - \bar{\alpha}_t)\mathsf{K}_t \cdot \mathrm{I}_{\mathsf{d}_y}\right),$$

where $\mathsf{K}_t := \sigma_{t+1}^2/(\sigma_{t+1}^2 + 1 - \alpha_t)$. For this procedure to target the posterior $\phi_0^y$, the weight function $\widetilde{\omega}_s$ is chosen as follows; we set $\widetilde{\omega}_{n-1}(x_n) := \int p_{n-1}(x_{n-1}|x_n)\overline{q}_{n-1|0}(\overline{x}_{n-1}|y)\mathrm{d}x_{n-1} = \mathcal{N}\left(\alpha_{n-1}^{1/2}y; \overline{\mathsf{m}}_n(x_n), \sigma_n^2 + 1 - \alpha_{n-1}\right)$ and for $t \in [1 : n - 2]$,

$$\widetilde{\omega}_t(x_{t+1}) := \frac{\int p_t(x_t|x_{t+1})\overline{q}_{t|0}(\overline{x}_t|y)\mathrm{d}x_t}{\overline{q}_{t+1|0}(\overline{x}_{t+1}|y)} = \frac{\mathcal{N}\left(\alpha_t^{1/2}y; \overline{\mathsf{m}}_{t+1}(x_{s+1}), (\sigma_{t+1}^2 + 1 - \alpha_t)\mathrm{I}_{\mathsf{d}_y}\right)}{\mathcal{N}\left(\alpha_{t+1}^{1/2}y; \overline{x}_{t+1}, (1 - \alpha_{t+1})\mathrm{I}_{\mathsf{d}_y}\right)}. \tag{2.4}$$

For the final step, we set $\widetilde{\omega}_0(x_1) := \overline{p}_0(y|\overline{x}_1)/\overline{q}_{1|0}(\overline{x}_1|y)$. The overall SMC algorithm targeting $\phi_0^y$ using the instrumental kernel equation 2.3 and weight function equation 2.4 is summarized in Algorithm 1. We now provide

---

**Algorithm 1:** MCGdiff ($\sigma = 0$)

---

**Input:** Number of particles $N$
**Output:** $\xi_0^{1:N}$
$\xi_n^{1:N} \overset{\text{i.i.d.}}{\sim} \mathcal{N}(\mathbf{0}_{d_x}, \mathrm{I}_{d_x})$;
// Operations involving index $i$ are repeated for each $i \in [1:N]$
**for** $s \leftarrow n-1 : 0$ **do**
 **if** $s = n-1$ **then**
  $\widetilde{\omega}_{n-1}(\xi_n^i) = \mathcal{N}\left(\bar{\alpha}_n^{1/2}y; \overline{\mathsf{m}}_n(\xi_n^i), 2 - \bar{\alpha}_n\right)$;
 **else**
  $\widetilde{\omega}_s(\xi_{s+1}^i) = \mathcal{N}\left(\bar{\alpha}_s^{1/2}y; \overline{\mathsf{m}}_{s+1}(\xi_{s+1}^i), \sigma_{s+1}^2 + 1 - \bar{\alpha}_s\right)/\mathcal{N}\left(\bar{\alpha}_{s+1}^{1/2}y; \overline{\xi}_{s+1}^i, 1 - \bar{\alpha}_{s+1}\right)$;
 $I_{s+1}^i \sim \mathrm{Cat}\left(\{\widetilde{\omega}_s(\xi_{s+1}^j)/\sum_{k=1}^N \widetilde{\omega}_s(\xi_{s+1}^k)\}_{j=1}^N\right), \quad \overline{z}_s^i \sim \mathcal{N}(\mathbf{0}_{d_y}, \mathrm{I}_{d_y}), \quad \underline{z}_s^i \sim \mathcal{N}(\mathbf{0}_{d_x - d_y}, \mathrm{I}_{d_x - d_y})$;
 $\overline{\xi}_s^i = \mathsf{K}_s \bar{\alpha}_s^{1/2} y + (1 - \mathsf{K}_s)\overline{\mathsf{m}}_{s+1}(\xi_{s+1}^{I_{s+1}^i}) + (1 - \alpha_s)^{1/2}\mathsf{K}_s^{1/2}\overline{z}_s^i, \quad \underline{\xi}_s^i = \underline{\mathsf{m}}_{s+1}(\xi_{s+1}^{I_{s+1}^i}) + \sigma_{s+1}\underline{z}_s^i$;
 Set $\xi_s^i = \overline{\xi}_s^i {}^\frown \underline{\xi}_s^i$;

---

a justification to Algorithm 1. Let $\{g_s^y\}_{s=1}^n$ be a sequence of positive functions with $g_n^y \equiv 1$. Consider the sequence of distributions $\{\phi_s^y\}_{s=1}^n$ defined as follows; $\phi_n^y(x_n) \propto g_n^y(x_n)\mathsf{p}_n(x_n)$ and for $t \in [1 : n-1]$

$$\phi_t^y(x_t) \propto \int g_{t+1}^y(x_{t+1})^{-1} g_t^y(x_t) p_t(x_t|x_{t+1})\phi_{t+1}^y(\mathrm{d}x_{t+1}). \tag{2.5}$$

By construction, the time $t$ marginal equation 2.5 is $\phi_t^y(x_t) \propto \mathsf{p}_t(x_t)g_t^y(x_t)$ for all $t \in [1 : n]$. Then, using $\phi_1^y$ and equation 2.2, we have that $\phi_0^y(\underline{x}_0) \propto \int g_1^y(x_1)^{-1}\overline{p}_0(y|\overline{x}_1)\underline{p}_0(\underline{x}_0|x_1)\phi_1^y(\mathrm{d}x_1)$.

The recursion equation 2.5 suggests a way of obtaining a particle approximation of $\phi_0^y$; by sequentially approximating each $\phi_t^y$ we can effectively derive a particle approximation of the posterior. To construct the intermediate particle approximations we use the framework of *auxiliary particle filters* (APF) (Pitt & Shephard, 1999). We focus on the case $g_t^y(x_t) = \overline{q}_{t|0}(\overline{x}_t|y)$ which corresponds to Algorithm 1. The initial particle approximation $\phi_n^y$ is obtained by drawing $N$ i.i.d. samples $\xi_n^{1:N}$ from $\mathsf{p}_n$ and setting $\phi_n^N = N^{-1}\sum_{i=1}^N \delta_{\xi_n^i}$ where $\delta_\xi$ is the Dirac mass at $\xi$. Assume that the empirical approximation of $\phi_{t+1}^y$ is $\phi_{t+1}^N = N^{-1}\sum_{i=1}^N \delta_{\xi_{t+1}^i}$, where $\xi_{t+1}^{1:N}$ are $N$ random variables. Substituting $\phi_{t+1}^N$ into the recursion equation 2.5 and introducing the instrumental kernel equation 2.3, we obtain the mixture

$$\widehat{\phi}_t^N(x_t) = \sum_{i=1}^N \widetilde{\omega}_t(\xi_{t+1}^i) p_t^y(x_t|\xi_{t+1}^i)/\sum_{j=1}^N \widetilde{\omega}_t(\xi_{t+1}^j). \tag{2.6}$$

Then, a particle approximation of equation 2.6 is obtained by sampling $N$ conditionally i.i.d. ancestor indices $I_{t+1}^{1:N} \overset{\text{i.i.d.}}{\sim} \mathrm{Cat}(\{\widetilde{\omega}_t(\xi_{t+1}^i)/\sum_{j=1}^N \widetilde{\omega}_t(\xi_{t+1}^j)\}_{i=1}^N)$, and then propagating each ancestor particle $\xi_{t+1}^{I_{t+1}^i}$ according to the instrumental kernel equation 2.3. The final particle approximation is given by $\phi_0^N = N^{-1}\sum_{i=1}^N \delta_{\underline{\xi}_0^i}$, where $\underline{\xi}_0^i \sim \underline{p}_0(\cdot|\xi_1^{I_1^i}), I_1^i \sim \mathrm{Cat}(\{\widetilde{\omega}_0(\xi_1^k)/\sum_{j=1}^N \widetilde{\omega}_0(\xi_1^j)\}_{k=1}^N)$. The sequence of distributions $\{\mathsf{p}_t\}_{t=0}^n$ approximating the marginals of the forward process initialized at $\mathsf{p}_0$ defines a path that bridges between $\mathsf{p}_n$ and the prior $\mathsf{p}_0$ such that the discrepancy between $\mathsf{p}_t$ and $\mathsf{p}_{t+1}$ is small. SMC samplers based on this path are robust to multi-modality and offer an interesting alternative to the geometric and tempering paths traditionally used in the SMC literature, see Dai et al. (2022). Our proposals $\phi_t^y(x_t) \propto \mathsf{p}_t(x_t)\overline{q}_{t|0}(\overline{x}_t|y)$ inherit the behavior of $\{\mathsf{p}_t\}_{t\in\mathbb{N}}$ and bridge the initial distribution $\phi_n^y$ and posterior $\phi_0^y$. Indeed, as $y$ is a noiseless observation of $X_0 \sim \mathsf{p}_0$, we may consider $\bar{\alpha}_t^{1/2}y + (1 - \bar{\alpha}_t)^{1/2}\varepsilon_t$, with $\varepsilon_t \sim \mathcal{N}(\mathbf{0}_{d_y}, \mathrm{I}_{d_y})$, as a noisy observation of $X_t \sim \mathsf{p}_t$ and thus, $\phi_t^y$ is the associated posterior. We illustrate this intuition by considering the following Gaussian mixture (GM) example. We assume that $\mathsf{p}_0(x_0) = \sum_{i=1}^M w_i \cdot \mathcal{N}(x_0; \mu_i, \mathrm{I}_{d_x})$ where $M > 1$ and $\{w_i\}_{i=1}^M$ are drawn uniformly on the simplex. The marginals of the forward process are available in closed form and are given by

| $t = 450$ | $t = 100$ | $t = 80$ | $t = 70$ | $t = 50$ | $t = 20$ | $t = 15$ | $t = 5$ |
|---|---|---|---|---|---|---|---|

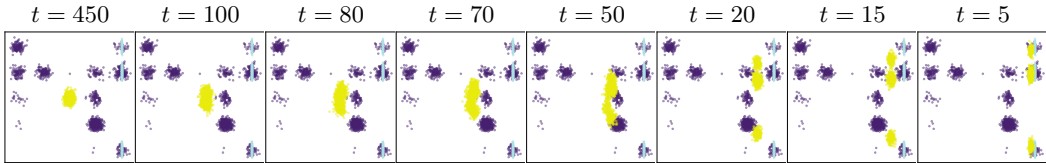

Figure 1: Display of samples from $\phi_t^y(x_t) \propto \mathsf{p}_t(x_t)\overline{q}_{t|0}(\overline{x}_t|y)$ for the GM prior. Samples from $\phi_t^y$ (yellow), those from the prior (purple) and those from the posterior $\phi_0^y$ (light blue) with $n = 500$.

$\mathsf{p}_t(x_t) = \sum_{i=1}^M w_i \cdot \mathcal{N}(x_t; \bar{\alpha}_t^{1/2}\mu_i, \mathrm{I}_{\mathsf{d}_x})$, which shows that the discrepancy between $\mathsf{p}_t$ and $\mathsf{p}_{t+1}$ is small as long as $\bar{\alpha}_t^{1/2} - \bar{\alpha}_{t+1}^{1/2}$ is close to 0. The posteriors $\{\phi_t^y\}_{t\in[0:n]}$ are also available in closed form and displayed in Figure 1, which illustrates that our choice of potentials ensures that the discrepancy between consecutive posteriors is small. The idea of using the forward diffused observation to guide the observed part of the state, as we do here through $\overline{q}_t(\overline{x}_t|y)$, has been exploited in prior works but in a different way. For instance, in Song et al. (2021c; 2022) the observed part of the state is directly replaced by the forward noisy observation and, as it has been noted Trippe et al. (2023), this introduces an irreducible bias. Instead, MCGdiff weights the backward process by the density of the forward one conditioned on $y$, resulting in a natural and consistent algorithm.

We now establish the convergence of MCGdiff with a general sequence of *potentials* $\{g_s^y\}_{s=1}^n$. We consider the following assumption on the sequence of potentials $\{g_t^y\}_{t=1}^n$.

(A1) $\sup_{x\in\mathbb{R}^{\mathsf{d}_x}} \overline{p}_0(y|x)/g_1^y(x) < \infty$ and $\sup_{x\in\mathbb{R}^{\mathsf{d}_x}} \int g_t^y(x_t)p_t(x_t|x)\mathrm{d}x_t/g_{t+1}^y(x) < \infty$ for all $t \in [1:n-1]$.

The following exponential deviation inequality is standard and is a direct application of (Douc et al., 2014, Theorem 10.17). In particular, it implies a $\mathcal{O}(1/\sqrt{N})$ bound on the mean squared error $\|\phi_0^N(h) - \phi_0^y(h)\|_2$.

**Proposition 2.1.** *Assume (A1). There exist constants $c_{1,n}, c_{2,n} \in (0,\infty)$ such that, for all $N \in \mathbb{N}$, $\varepsilon > 0$ and bounded function $h : \mathbb{R}^{\mathsf{d}_x} \mapsto \mathbb{R}$, $\mathbb{P}\left[\left|\phi_0^N(h) - \phi_0^y(h)\right| \geq \varepsilon\right] \leq c_{1,n}\exp(-c_{2,n}N\varepsilon^2/|h|_\infty^2)$ where $|h|_\infty := \sup_{x\in\mathbb{R}^{\mathsf{d}_x}} |h(x)|$.*

We also furnish our estimator with an explicit non-asymptotic bound on its bias. Define $\Phi_0^N = \mathbb{E}[\phi_0^N]$ where $\phi_0^N = N^{-1}\sum_{i=1}^N \delta_{\xi_0^i}$ is the particle approximation produced by Algorithm 1 and the expectation is with respect to the law of $(\xi_{0:n}^{1:N}, I_{1:n}^{1:N})$. Define for all $t \in [1:n]$, $\phi_t^\star(x_t) \propto \mathsf{p}_t(x_t) \int \delta_y(\mathrm{d}\overline{x}_0)p_{0|t}(x_0|x_t)\mathrm{d}\underline{x}_0$, where $p_{0|t}(x_0|x_t) := \int \left\{\prod_{s=0}^{t-1} p_s(x_s|x_{s+1})\right\} \mathrm{d}x_{1:t-1}$.

**Proposition 2.2.** *It holds that*

$$\mathsf{KL}(\phi_0^y \parallel \Phi_0^N) \leq \mathsf{C}_{0:n}^y(N-1)^{-1} + \mathsf{D}_{0:n}^y N^{-2}, \tag{2.7}$$

*where $\mathsf{D}_{0:n}^y > 0$, $\mathsf{C}_{0:n}^y := \sum_{t=1}^n \int \frac{\mathcal{Z}_t/\mathcal{Z}_0}{g_t^y(z_t)} \left\{\int \delta_y(\mathrm{d}\overline{x}_0)p_{0|t}(x_0|z_t)\mathrm{d}\underline{x}_0\right\} \phi_t^\star(\mathrm{d}z_t)$ and $\mathcal{Z}_t := \int g_t^y(x_t)\mathsf{p}_t(\mathrm{d}x_t)$ for all $t \in [1:n]$ and $\mathcal{Z}_0 := \int \delta_y(\mathrm{d}\overline{x}_0)\mathsf{p}_0(x_0)\mathrm{d}\underline{x}_0$. If furthermore (A1) holds then both $\mathsf{C}_{0:n}^y$ and $\mathsf{D}_{0:n}^y$ are finite.*

The proof of Proposition 2.2 is postponed to Appendix B.1. (A1) is an assumption on the equivalent of the weights $\{\widetilde{\omega}_t\}_{t=0}^n$ with a general sequence of potentials $\{g_t^y\}_{t=1}^n$ and is not restrictive as it can be satisfied by setting for example $g_s^y(x_s) = \overline{q}_{s|0}(\overline{x}_s|y) + \delta$ where $\delta > 0$. The resulting algorithm is then only a slight modification of the one described above, see Appendix B.1 for more details. It is also worth noting that Proposition 2.2 combined with Pinsker's inequality implies that the bias of MCGdiff goes to 0 with the number of particle samples $N$ for fixed $n$. We have chosen to present a bound in Kullback–Leibler (KL) divergence, inspired by Andrieu et al. (2018); Huggins & Roy (2019), as it allows an explicit dependence on the modeling choice $\{g_s^y\}_{s=1}^n$, see Lemma B.2. Finally, unlike the theoretical guarantees established for SMCdiff in Trippe et al. (2023), proving

the asymptotic exactness of our methodology w.r.t. to the generative model posterior does not require having $\mathsf{p}_{s+1}(x_{s+1})p_s(x_s|x_{s+1}) = \mathsf{p}_s(x_s)q_{s+1}(x_{s+1}|x_s)$ for all $s \in [0 : n-1]$, which does not hold in practice.

**Noisy case.** We consider the case $\sigma_y > 0$. The posterior density is given by $\phi_0^y(x_0) \propto g_0^y(\overline{x}_0)\mathsf{p}_0(x_0)$, where $g_0^y(x_0) := \mathcal{N}(y; \overline{x}_0, \sigma_y^2 \mathrm{I}_{\mathsf{d}_y})$. In what follows, assume that there exists $\tau \in [1 : n]$ such that $\sigma^2 = (1 - \bar{\alpha}_\tau)/\bar{\alpha}_\tau$. We denote $\tilde{y}_\tau = \bar{\alpha}_\tau^{1/2} y$. We can then write that

$$g_0^y(\overline{x}_0) = \bar{\alpha}_\tau^{1/2} \cdot \mathcal{N}(\tilde{y}_\tau; \bar{\alpha}_\tau^{1/2} x_0, (1 - \bar{\alpha}_\tau) \cdot \mathrm{I}_{\mathsf{d}_y}) = \bar{\alpha}_\tau^{1/2} \cdot \overline{q}_{\tau|0}(\tilde{y}_\tau|\overline{x}_0), \tag{2.8}$$

which hints that the likelihood function $g_0^y$ is closely related to the forward process equation 1.1. We may then write the posterior $\phi_0^y(x_0)$ as $\phi_0^y(x_0) \propto \overline{q}_{\tau|0}(\tilde{y}_\tau|\overline{x}_0)\mathsf{p}_0(x_0) \propto \int \delta_{\tilde{y}_\tau}(\mathrm{d}\overline{x}_\tau)q_{\tau|0}(x_\tau|x_0)\mathsf{p}_0(x_0)\mathrm{d}\underline{x}_\tau$. Next, assume that the forward process equation 1.1 is the reverse of the backward one equation 1.3, i.e. that

$$\mathsf{p}_t(x_t)q_{t+1}(x_{t+1}|x_t) = \mathsf{p}_{t+1}(x_{t+1})p_t(x_t|x_{t+1}), \quad \forall t \in [0 : n-1]. \tag{2.9}$$

This is similar to the assumption made in SMCdiff Trippe et al. (2023). Then, it is easily seen that it implies $\mathsf{p}_0(x_0)q_{\tau|0}(x_\tau|x_0) = \mathsf{p}_\tau(x_\tau)p_{0|\tau}(x_0|x_\tau)$ and thus

$$\phi_0^y(x_0) = \int p_{0|\tau}(x_0|x_\tau)\delta_{\tilde{y}_\tau}(\mathrm{d}\overline{x}_\tau)\mathsf{p}_\tau(x_\tau)\mathrm{d}\underline{x}_\tau \bigg/ \int \delta_{\tilde{y}_\tau}(\mathrm{d}\overline{z}_\tau)\mathsf{p}_\tau(z_\tau)\mathrm{d}\underline{z}_\tau = \int p_{0|\tau}(x_0|\tilde{y}_\tau^\frown \underline{x}_\tau)\phi_\tau^{\tilde{y}_\tau}(\mathrm{d}\underline{x}_\tau), \tag{2.10}$$

where $\phi_\tau^{\tilde{y}_\tau}(\underline{x}_\tau) \propto \mathsf{p}_\tau(\tilde{y}_\tau^\frown \underline{x}_\tau)$. equation 2.10 highlights that solving the inverse problem equation 2.1 with $\sigma_y > 0$ is equivalent to solving an inverse problem on the intermediate state $X_\tau \sim \mathsf{p}_\tau$ with *noiseless* observation $\tilde{y}_\tau$ of the $\mathsf{d}_y$ top coordinates and then propagating the resulting posterior back to time 0 with the backward kernel $p_{0|\tau}$. The assumption equation 2.9 does not always holds in realistic settings.Therefore, while equation 2.10 also holds only approximately in practice, we can still use it as inspiration for designing potentials when the assumption is not valid. Consider then $\{g_t^y\}_{t=\tau}^n$ and sequence of probability measures $\{\phi_t^y\}_{t=\tau}^n$ defined for all $t \in [\tau : n]$ as $\phi_t^y(x_t) \propto g_t^y(x_t)\mathsf{p}_t(x_t)$, where $g_t^y(x_t) := \mathcal{N}(x_t; \bar{\alpha}_t^{1/2}y, (1 - (1-\kappa)\bar{\alpha}_t/\bar{\alpha}_\tau)\mathrm{I}_{\mathsf{d}_y})$, $\kappa \geq 0$. In the case of $\kappa = 0$, we have $g_t^y(x_t) = \overline{q}_{t|\tau}(\overline{x}_t|\tilde{y}_\tau)$ for $t \in [\tau+1 : n]$ and $\phi_\tau^y = \phi_\tau^{\tilde{y}_\tau}$. The recursion equation 2.5 holds for $t \in [\tau : n]$ and assuming $\kappa > 0$, we find that $\phi_0^y(x_0) \propto g_0^y(x_0) \int g_\tau^y(x_\tau)^{-1}p_{0|\tau}(x_0|x_\tau)\phi_\tau^y(\mathrm{d}x_\tau)$, which resembles the recursion equation 2.10. In practice we take $\kappa$ to be small in order to mimic the Dirac delta mass at $\overline{x}_\tau$ in equation 2.10. Having a particle approximation $\phi_\tau^N = N^{-1}\sum_{i=1}^N \delta_{\xi_\tau^i}$ of $\phi_\tau^y$ by adapting Algorithm 1, we estimate $\phi_0^y$ with $\phi_0^N = \sum_{i=1}^N \omega_0^i \delta_{\xi_0^i}$ where $\xi_0^i \sim p_{0|\tau}(\cdot|\xi_\tau^i)$ and $\omega_0^i \propto g_0^y(\xi_0^i)/g_\tau^y(\xi_\tau^{I_\tau^i})$. In the next section we extend this methodology to general linear Gaussian observation models. Finally, equation 2.10 allows us to extend SMCdiff to handle noisy inverse problems in a principled manner which is detailed in Appendix A.

## 2.1 EXTENSION TO GENERAL LINEAR INVERSE PROBLEMS

Consider $Y = \mathrm{A}X + \sigma_y \varepsilon$ where $\mathrm{A} \in \mathbb{R}^{\mathsf{d}_y \times \mathsf{d}_x}$, $\varepsilon \sim \mathcal{N}(0_{\mathsf{d}_y}, \mathrm{I}_{\mathsf{d}_y})$ and $\sigma_y \geq 0$ and the singular value decomposition (SVD) $\mathrm{A} = \mathrm{U}\mathrm{S}\overline{\mathrm{V}}^T$, where $\overline{\mathrm{V}} \in \mathbb{R}^{\mathsf{d}_x \times \mathsf{d}_y}$, $\mathrm{U} \in \mathbb{R}^{\mathsf{d}_y \times \mathsf{d}_y}$ are two orthonormal matrices, and $\mathrm{S} \in \mathbb{R}^{\mathsf{d}_y \times \mathsf{d}_y}$ is diagonal. For simplicity, it is assumed that the singular values satisfy $s_1 > \cdots > s_{\mathsf{d}_y} > 0$. Set $\mathsf{b} = \mathsf{d}_x - \mathsf{d}_y$. Let $\underline{\mathrm{V}} \in \mathbb{R}^{\mathsf{d}_x \times \mathsf{b}}$ be an orthonormal matrix of which the columns complete those of $\overline{\mathrm{V}}$ into an orthonormal basis of $\mathbb{R}^{\mathsf{d}_x}$, i.e. $\underline{\mathrm{V}}^T\underline{\mathrm{V}} = \mathrm{I}_{\mathsf{b}}$ and $\underline{\mathrm{V}}^T\overline{\mathrm{V}} = 0_{\mathsf{b},\mathsf{d}_y}$. We define $\mathrm{V} = [\overline{\mathrm{V}}, \underline{\mathrm{V}}] \in \mathbb{R}^{\mathsf{d}_x \times \mathsf{d}_x}$. In what follows, for a given $\mathbf{x} \in \mathbb{R}^{\mathsf{d}_x}$ we write $\overline{\mathbf{x}} \in \mathbb{R}^{\mathsf{d}_y}$ for its top $\mathsf{d}_y$ coordinates and $\underline{\mathbf{x}} \in \mathbb{R}^{\mathsf{b}}$ for the remaining coordinates. Setting $\mathbf{X} := \mathrm{V}^T X$ and $\mathbf{Y} := \mathrm{S}^{-1}\mathrm{U}^T Y$ and multiplying the measurement equation by $\mathrm{S}^{-1}\mathrm{U}^T$ yields

$$\mathbf{Y} = \overline{\mathbf{X}} + \sigma_y \mathrm{S}^{-1}\tilde{\varepsilon}, \quad \tilde{\varepsilon} \sim \mathcal{N}(0, \mathrm{I}_{\mathsf{d}_y}).$$

In this section, we focus on solving this linear inverse problem in the orthonormal basis defined by V using the methodology developed in the previous sections. This prompts us to define the diffusion based generative model in this basis. As V is an orthonormal matrix, the law of $\mathbf{X}_0 = \mathrm{V}^T X_0$ is $\mathfrak{p}_0(\mathbf{x}_0) := \mathsf{p}_0(\mathrm{V}\mathbf{x}_0)$. By definition of $\mathfrak{p}_0$

and the fact that $\|V\mathbf{x}\|_2 = \|\mathbf{x}\|_2$ for all $\mathbf{x} \in \mathbb{R}^{\mathsf{d}_x}$ we have that

$$\mathfrak{p}_0(\mathbf{x}_0) = \int p_0(V\mathbf{x}_0|x_1) \left\{ \prod_{s=1}^{n-1} p_s(\mathrm{d}x_s|x_{s+1}) \right\} \mathfrak{p}_n(\mathrm{d}x_n) = \int \lambda_0(\mathbf{x}_0|\mathbf{x}_1) \left\{ \prod_{s=1}^{n-1} \lambda_s(\mathrm{d}\mathbf{x}_s|\mathbf{x}_{s+1}) \right\} \mathfrak{p}_n(\mathrm{d}\mathbf{x}_n)$$

where for all $s \in [1:n]$, $\lambda_{s-1}(\mathbf{x}_{s-1}|\mathbf{x}_s) := \mathcal{N}(\mathbf{x}_{s-1}; \mathfrak{m}_s(\mathbf{x}_s), \sigma_s^2 I_{\mathsf{d}_x})$, where $\mathfrak{m}_s(\mathbf{x}_s) := V^T m_s(V\mathbf{x}_s)$. The transition kernels $\{\lambda_s\}_{s=0}^{n-1}$ define a diffusion based model in the basis V. We write $\overline{\mathfrak{m}}_s(\mathbf{x}_s)$ for the first $\mathsf{d}_{\mathbf{y}}$ coordinates of $\mathfrak{m}_s(\mathbf{x}_s)$ and $\underline{\mathfrak{m}}_s(\mathbf{x}_s)$ the last b coordinates and denote by $\mathfrak{p}_s$ the time $s$ marginal of the backward process.

**Noiseless.** In this case the target posterior is $\phi_0^{\mathbf{y}}(\mathbf{x}_0) \propto \mathfrak{p}_0(\mathbf{y}^\frown \underline{\mathbf{x}}_0)$. The extension of algorithm 1 is straight forward; it is enough to replace $y$ with $\mathbf{y}$ ($= S^{-1}U^T y$) and the backward kernels $\{p_t\}_{t=0}^{n-1}$ with $\{\lambda_t\}_{t=0}^{n-1}$.

**Noisy.** The posterior density is then $\phi_0^{\mathbf{y}}(\mathbf{x}_0) \propto g_0^{\mathbf{y}}(\overline{\mathbf{x}}_0)\mathfrak{p}_0(\mathbf{x}_0)$, where $g_0^{\mathbf{y}}(\overline{\mathbf{x}}_0) = \prod_{i=1}^{\mathsf{d}_{\mathbf{y}}} \mathcal{N}(\mathbf{y}[i]; \overline{\mathbf{x}}_0[i], (\sigma_y/s_i)^2)$. As in Section 2, assume that there exists $\{\tau_i\}_{i=1}^{\mathsf{d}_{\mathbf{y}}} \subset [1:n]$ such that $\bar{\alpha}_{\tau_i}\sigma_y^2 = (1 - \bar{\alpha}_{\tau_i})s_i^2$ and define for all $i \in [1:\mathsf{d}_{\mathbf{y}}]$, $\tilde{\mathbf{y}}_i := \bar{\alpha}_{\tau_i}^{1/2}\mathbf{y}[i]$. Then we can write the potential $g_0^{\mathbf{y}}$ in a similar fashion to equation 2.8 as the product of forward processes from time 0 to each time step $\tau_i$, i.e. $g_0^{\mathbf{y}}(\mathbf{x}_0) = \prod_{i=1}^{\mathsf{d}_{\mathbf{y}}} \bar{\alpha}_{\tau_i}^{1/2}\mathcal{N}(\tilde{\mathbf{y}}_i; \bar{\alpha}_{\tau_i}^{1/2}\mathbf{x}_0[i], (1 - \bar{\alpha}_{\tau_i}))$. Writing the potential this way allows us to generalize equation 2.10 as follows. Denote for $\ell \in [1:\mathsf{d}_x]$, $\mathbf{x}^{\backslash \ell} \in \mathbb{R}^{\mathsf{d}_x - 1}$ the vector $\mathbf{x}$ with its $\ell$-th coordinate removed. Define

$$\phi_{\tau_1:n}^{\tilde{\mathbf{y}}}(\mathrm{d}\mathbf{x}_{\tau_1:n}) \propto \left\{ \prod_{i=1}^{\mathsf{d}_{\mathbf{y}}-1} \lambda_{\tau_i|\tau_{i+1}}(\mathbf{x}_{\tau_i}|\mathbf{x}_{\tau_{i+1}})\delta_{\tilde{\mathbf{y}}_i}(\mathrm{d}\mathbf{x}_{\tau_i}[i])\mathrm{d}\mathbf{x}_{\tau_i}^{\backslash i} \right\} \mathfrak{p}_{\tau_{\mathsf{d}_{\mathbf{y}}}}(\mathbf{x}_{\tau_{\mathsf{d}_{\mathbf{y}}}})\delta_{\tilde{\mathbf{y}}_{\mathsf{d}_{\mathbf{y}}}}(\mathrm{d}\mathbf{x}_{\tau_{\mathsf{d}_{\mathbf{y}}}}[\mathsf{d}_{\mathbf{y}}])\mathrm{d}\mathbf{x}_{\tau_{\mathsf{d}_{\mathbf{y}}}}^{\backslash \mathsf{d}_{\mathbf{y}}} ,$$

which corresponds to the posterior of a noiseless inverse problem on the joint states $\mathbf{X}_{\tau_1:n} \sim \mathfrak{p}_{\tau_1:n}$ with noiseless observations $\tilde{\mathbf{y}}_{\tau_i}$ of $\mathbf{X}_{\tau_i}[i]$ for all $i \in [1:\mathsf{d}_{\mathbf{y}}]$.

**Proposition 2.3.** *Assume that* $\mathfrak{p}_{s+1}(\mathbf{x}_{s+1})\lambda_s(\mathbf{x}_s|\mathbf{x}_{s+1}) = \mathfrak{p}_s(\mathbf{x}_s)q_{s+1}(\mathbf{x}_{s+1}|\mathbf{x}_s)$ *for all* $s \in [0:n-1]$. *Then it holds that* $\phi_0^{\mathbf{y}}(\mathbf{x}_0) \propto \int \lambda_{0|\tau_1}(\mathbf{x}_0|\mathbf{x}_{\tau_1})\phi_{\tau_1:n}^{\tilde{\mathbf{y}}}(\mathrm{d}\mathbf{x}_{\tau_1:n})$.

The proof of Proposition 2.3 is given in Appendix B.2. We have shown that sampling from $\phi_0^{\mathbf{y}}$ is equivalent to sampling from $\phi_{\tau_1:n}^{\tilde{\mathbf{y}}}$ then propagating the final state $\mathbf{X}_{\tau_1}$ to time 0 according to $\lambda_{0|\tau_1}$. Therefore, as in equation 2.8, we define $\{g_t^y\}_{t=\tau_1}^n$ and $\{\phi_t^{\mathbf{y}}\}_{t=\tau_1}^n$ for all $t \in [\tau_1:n]$ by $\phi_t^{\mathbf{y}}(\mathbf{x}_t) \propto g_t^{\mathbf{y}}(\mathbf{x}_t)p_t(\mathbf{x}_t)$ and $g_t^{\mathbf{y}}(\mathbf{x}_t) := \prod_{i=1}^{\tau(t)} \mathcal{N}\left(\mathbf{x}_t; \bar{\alpha}_t^{1/2}\mathbf{y}_i, 1 - (1-\kappa)\bar{\alpha}_t/\bar{\alpha}_{\tau_i}\right)$, $\kappa > 0$. We obtain a particle approximation of $\phi_{\tau_1}^{\mathbf{y}}$ using a particle filter with proposal kernel and weight function $\lambda_t^{\mathbf{y}}(\mathbf{x}_t|\mathbf{x}_{t+1}) \propto g_t^{\mathbf{y}}(\mathbf{x}_t)p_t(\mathbf{x}_t|\mathbf{x}_{t+1})$, $\widetilde{\omega}_t(\mathbf{x}_{t+1}) = \int g_t^{\mathbf{y}}(\mathbf{x}_t)p_t(\mathrm{d}\mathbf{x}_t|\mathbf{x}_{t+1})/g_{t+1}^{\mathbf{y}}(\mathbf{x}_{t+1})$, which are both available in closed form.

## 3 NUMERICS

A prerequisite for quantitative evaluation in ill-posed inverse problems in a Bayesian setting is to have access to samples of the posterior distribution. This generally requires having at least an unnormalized proxy of the posterior density, so that one can run MCMC samplers such as the No U-turn sampler (NUTS) Hoffman & Gelman (2011). Therefore, this section focus on mixture models of two types of basis distribution, the Gaussian and the Funnel distributions. We then present a brief illustration of MCGdiff on image data. However, in this setting, the actual posterior distribution is unknown and the main goal is to explore the potentially multimodal posterior distribution, which makes a comparison with a "real image" meaningless. Therefore, metrics such as Fréchet Inception Distance (FID) and LPIPS score, which require comparison to a ground truth, are not useful for evaluating Bayesian reconstruction methods in such settings.[1]

**Mixture Models.** We refer to the Funnel mixture prior as FM prior (see Appendix B.3). For GM prior, we consider a mixture of 25 components with known means and variances. For FM prior, we consider a mixture of 20 components consisting of rotated and translated funnel distributions. For a given pair $(\mathsf{d}_x, \mathsf{d}_{\mathbf{y}})$, we sample a prior distribution by randomly sampling the weights of the mixture and for the FM case the translation and rotation of each component. We then randomly sample measurement models $(y, A, \sigma_y) \in \mathbb{R}^{\mathsf{d}_{\mathbf{y}}} \times \mathbb{R}^{\mathsf{d}_{\mathbf{y}} \times \mathsf{d}_x} \times [0, 1]$. For each pair of prior distribution and measurement model, we generate $10^4$ samples from MCGdiff, DPS, DDRM, RNVP, and from the posterior either analytically (GM) or using NUTS (FM). We calculate for each algorithm the sliced Wasserstein (SW) distance between the resulting samples and the posterior samples. Table 1 shows the CLT

---

[1]The code for the experiments is available at https://github.com/gabrielvc/mcg_diff.

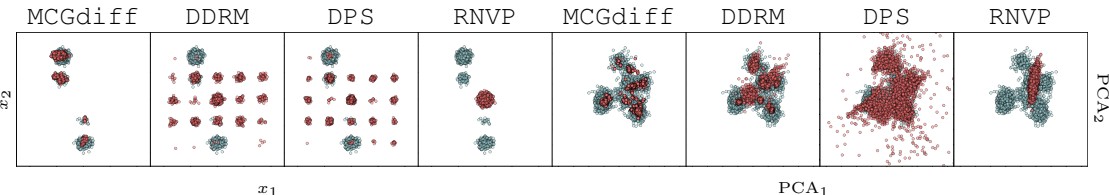

Figure 2: The first and last four columns correspond respectively to GM with $(\mathsf{d}_x, \mathsf{d}_\mathbf{y}) = (800, 1)$ and FM with $(\mathsf{d}_x, \mathsf{d}_\mathbf{y}) = (10, 1)$. The blue and red dots represent respectively samples from the exact posterior and those generated by each of the algorithms used (names on top).

| $d$ | $d_y$ | MCGdiff | DDRM | DPS | RNVP | $d$ | $d_y$ | MCGdiff | DDRM | DPS | RNVP |
|---|---|---|---|---|---|---|---|---|---|---|---|
| 80 | 1 | **1.39 ± 0.45** | 5.64 ± 1.10 | 4.98 ± 1.14 | 6.86 ± 0.88 | 6 | 1 | **1.95 ± 0.43** | 4.20 ± 0.78 | 5.43 ± 1.05 | 6.16 ± 0.65 |
| 80 | 2 | **0.67 ± 0.24** | 7.07 ± 1.35 | 5.10 ± 1.23 | 7.79 ± 1.50 | 6 | 3 | **0.73 ± 0.33** | 2.20 ± 0.67 | 3.47 ± 0.78 | 4.70 ± 0.90 |
| 80 | 4 | **0.28 ± 0.14** | 7.81 ± 1.48 | 4.28 ± 1.26 | 7.95 ± 1.61 | 6 | 5 | **0.41 ± 0.12** | 0.91 ± 0.43 | 2.07 ± 0.63 | 3.52 ± 0.93 |
| 800 | 1 | **2.40 ± 1.00** | 7.44 ± 1.15 | 6.49 ± 1.16 | 7.74 ± 1.34 | 10 | 1 | **2.45 ± 0.42** | 3.82 ± 0.64 | 4.30 ± 0.91 | 6.04 ± 0.38 |
| 800 | 2 | **1.31 ± 0.60** | 8.95 ± 1.12 | 6.88 ± 1.01 | 8.75 ± 1.02 | 10 | 3 | **1.07 ± 0.26** | 4.94 ± 0.87 | 5.38 ± 0.84 | 5.91 ± 0.64 |
| 800 | 4 | **0.47 ± 0.19** | 8.39 ± 1.48 | 5.51 ± 1.18 | 7.81 ± 1.63 | 10 | 5 | **0.71 ± 0.12** | 2.32 ± 0.74 | 3.74 ± 0.77 | 5.11 ± 0.69 |

Table 1: Sliced Wasserstein for the GM (left) and FM (right) case.

95% confidence intervals obtained over 20 seeds. Figure 2 illustrates the samples for the different algorithms for a given seed. We see that MCGdiff outperforms all the other algorithms in each setting tested. The complete details of the numerical experiments are available in Appendix B.3 as well as additional visualisations.

**Image datasets.** Figure 3 shows samples of MCGdiff in different datasets (Celeb, Churches, Bedroom and Flowers) for different inverse problems, namely Inpaiting (Inp), super resolution (SR), Gaussian 2D deblur (G2Deb) and Colorization (Col). Visual comparison with competing algorithms and different datasets are shown in Appendix B.3 as well as numerical details concerning Figure 3.

## 4 CONCLUSION

In this paper, we present MCGdiff a novel method for solving Bayesian linear Gaussian inverse problems with SGM priors. We show that MCGdiff is theoretically grounded and provided numerical experiments that reflect the adequacy of MCGdiff in a Bayesian framework, as opposed to recent works. This difference is of the uttermost importance when the relevance of the generated samples is hard to verify, as in safety critical applications. MCGdiff is a first step towards robust approaches for addressing the challenges of Bayesian linear inverse problems with SGM priors.

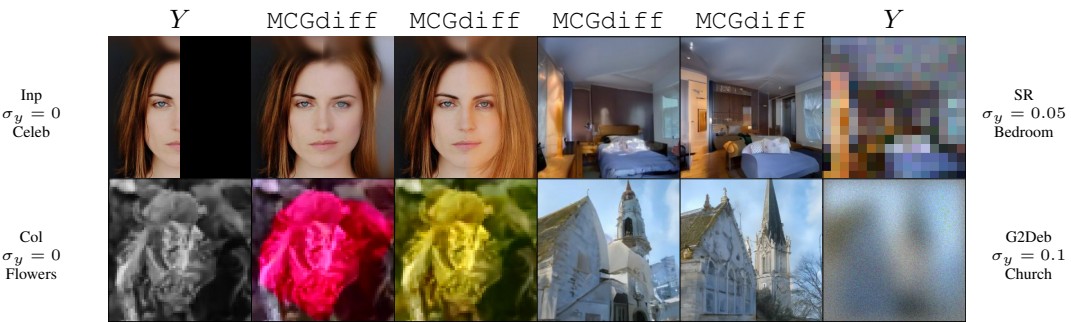

Figure 3: Illustration of the samples of MCGdiff for different datasets and different inverse problems.

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

## A   SMCDIFF EXTENSION

The identity equation 2.10 allows us to extend SMCdiff Trippe et al. (2023) to handle noisy inverse problems as we now show. We have that

$$\phi_\tau^{\tilde{y}_\tau}(\underline{x}_\tau) = \frac{\int p_\tau(\tilde{y}_\tau^\frown \underline{x}_\tau | x_{\tau+1}) \left\{ \prod_{s=\tau+1}^{n-1} p_s(\mathrm{d}x_s | x_{s+1}) \right\} \mathsf{p}_n(\mathrm{d}x_n)}{\int \mathsf{p}_\tau(\tilde{y}_\tau^\frown \underline{z}_\tau)\mathrm{d}\underline{z}_\tau}$$

$$= \int b_{\tau:n}^{\tilde{y}_\tau}(\underline{x}_{\tau:n} | \overline{x}_{\tau+1:n}) f_{\tau+1:n}^{\tilde{y}_\tau}(\mathrm{d}\overline{x}_{\tau+1:n})\mathrm{d}\underline{x}_{\tau+1:n} \,,$$

where

$$b_{\tau:n}(\underline{x}_{\tau:n} | \overline{x}_{\tau+1:n}) = \frac{p_\tau(\tilde{y}_\tau^\frown \underline{x}_\tau | x_{\tau+1}) \left\{ \prod_{s=\tau+1}^{n-1} \underline{p}_s(\underline{x}_s | x_{s+1}) \overline{p}_s(\overline{x}_s | x_{s+1}) \right\} \underline{\mathsf{p}}_n(\underline{x}_n)}{\mathsf{L}_{\tau:n}^{\tilde{y}_\tau}(\overline{x}_{\tau+1:n})} \,,$$

$$f_{\tau+1:n}^{\tilde{y}_\tau}(\overline{x}_{\tau+1:n}) = \frac{\mathsf{L}_{\tau:n}^{\tilde{y}_\tau}(\overline{x}_{\tau+1:n})}{\int \mathsf{p}_\tau(\tilde{y}_\tau^\frown \underline{z}_\tau)\mathrm{d}\underline{z}_\tau} \,,$$

and

$$\mathsf{L}_{\tau:n}^{\tilde{y}_\tau}(\overline{x}_{\tau+1:n}) = \int p_\tau(\tilde{y}_\tau^\frown \underline{z}_\tau | \overline{x}_{\tau+1}^\frown \underline{z}_{\tau+1}) \left\{ \prod_{s=\tau+1}^{n-1} \underline{p}_s(\mathrm{d}\underline{z}_s | \overline{x}_{s+1}^\frown \underline{z}_{s+1}) \overline{p}_s(\overline{x}_s | \overline{x}_{s+1}^\frown \underline{z}_{s+1}) \right\} \underline{\mathsf{p}}_n(\mathrm{d}\underline{z}_n) \,.$$

Next, equation 2.9 implies that

$$\int \mathsf{p}_{s+1}(\overline{x}_{s+1}^\frown \underline{z}_{s+1}) \underline{p}_s(\mathrm{d}\underline{z}_s | \overline{x}_{s+1}^\frown \underline{z}_{s+1}) \overline{p}_s(\overline{x}_s | \overline{x}_{s+1}^\frown \underline{z}_{s+1})\mathrm{d}\underline{z}_{s:s+1} =$$

$$\int \mathsf{p}_s(\overline{x}_s^\frown \underline{z}_s) \overline{q}_{s+1}(\overline{x}_{s+1} | \overline{x}_s) \underline{q}_{s+1}(\underline{z}_{s+1} | \underline{z}_s)\mathrm{d}\underline{z}_{s:s+1} \,,$$

and applied repeatedly, we find that

$$\mathsf{L}^{\tilde{y}_\tau}(\overline{x}_{\tau+1:n}) = \int \mathsf{p}_\tau(\tilde{y}_\tau^\frown \underline{x}_\tau)\mathrm{d}\underline{x}_\tau \cdot \int \delta_{\tilde{y}_\tau}(\mathrm{d}\overline{x}_\tau) \prod_{s=\tau+1}^{n} \overline{q}_s(\overline{x}_s | \overline{x}_{s-1}) \,.$$

and thus, $f_{\tau:n}^{\tilde{y}_\tau}(\overline{x}_{\tau+1:n}) = \int \delta_{\tilde{y}_\tau}(\mathrm{d}\overline{x}_\tau) \prod_{s=\tau+1}^{n} \overline{q}_s(\overline{x}_s | \overline{x}_{s-1})$. In order to approximate $\phi_\tau^{\tilde{y}_\tau}$ we first diffuse the noised observation up to time $n$, resulting in $\overline{x}_{\tau+1:n}$, and then estimate $b_{\tau+1:n}^{\tilde{y}_\tau}(\cdot | \overline{x}_{\tau+1:n})$ using a particle filter with $\underline{p}_s(\underline{x}_s | x_{s+1})$ as transition kernel at step $s \in [\tau + 1 : n]$ and $g_s : \underline{z}_s \mapsto \overline{p}_{s-1}(\overline{x}_{s-1} | \overline{x}_s^\frown \underline{z}_s)$ as potential, similarly to SMCdiff.

## B   PROOFS

### B.1   PROOF OF PROPOSITION 2.2

PRELIMINARY DEFINITIONS.

We preface the proof with notations and definitions of a few quantities that will be used throughout.

For a probability measure $\mu$ and $f$ a bounded measurable function, we write $\mu(f) := \int f(x)\mu(\mathrm{d}x)$ the expectation of $f$ under $\mu$ and if $K(\mathrm{d}x|z)$ is a transition kernel we write $K(f)(z) := \int f(x)K(\mathrm{d}x|z)$.

Define the *smoothing* distribution

$$\phi_{0:n}^y(\mathrm{d}x_{0:n}) \propto \delta_y(\mathrm{d}\overline{x}_0)\mathsf{p}_{0:n}(x_{0:n})\mathrm{d}\underline{x}_0\mathrm{d}x_{1:n} \,, \tag{B.1}$$

which admits the posterior $\phi_0^y$ as time 0 marginal. Its particle estimate known as the *poor man smoother* is given by

$$\phi_{0:n}^N(\mathrm{d}x_{0:n}) = N^{-1} \sum_{k_{0:n}\in[1:N]^{n+1}} \delta_{y\frown\underline{\xi}_0^{k_0}}(\mathrm{d}x_0) \prod_{s=1}^n \mathbb{1}\{k_s = I_s^{k_{s-1}}\}\delta_{\xi_s^{k_s}}(\mathrm{d}x_s). \tag{B.2}$$

We also let $\Phi_{0:n}^N$ be the probability measure defined for any $B \in \mathcal{B}(\mathbb{R}^{\mathsf{d}_x})^{\otimes n+1}$ by

$$\Phi_{0:n}^N(B) = \mathbb{E}\big[\phi_{0:n}^N(B)\big],$$

where the expectation is with respect to the probability measure

$$P_{0:n}^N\big(\mathrm{d}(x_{0:n}^{1:N}, a_{1:n}^{1:N})\big) = \prod_{i=1}^N p_n^y(\mathrm{d}x_n^i) \prod_{\ell=2}^n \left\{ \prod_{j=1}^N \sum_{k=1}^N \omega_{\ell-1}^k \delta_k(\mathrm{d}a_\ell^j) p_{\ell-1}^y(\mathrm{d}x_{\ell-1}^j|x_\ell^{a_\ell^j}) \right\}$$

$$\times \prod_{j=1}^N \sum_{k=1}^N \omega_0^k \delta_k(\mathrm{d}a_1^j) \underline{p}_0^y(\mathrm{d}\underline{x}_0^j|x_1^{a_1^j}) \delta_y(\mathrm{d}\overline{x}_0^j), \quad \text{(B.3)}$$

where $\omega_t^i := \widetilde{\omega}_t(\xi_{t+1}^i)/\sum_{j=1}^N \widetilde{\omega}_t(\xi_{t+1}^j)$ and which corresponds to the joint law of all the random variables generated by Algorithm 1. It then follows by definition that for any $C \in \mathcal{B}(\mathbb{R}^{\mathsf{d}_x})$,

$$\int \Phi_{0:n}^N(\mathrm{d}z_{0:n})\mathbb{1}_C(z_0) = \mathbb{E}\left[\int \phi_{0:n}^N(\mathrm{d}z_{0:n})\mathbb{1}_C(z_0)\right] = \mathbb{E}[\phi_0^N(C)] = \Phi_0^N(C).$$

Define also the law of the *conditional* particle cloud

$$\mathbf{P}^N\big(\mathrm{d}(x_{0:n}^{1:N}, a_{1:n}^{1:N})\big|z_{0:n}\big) = \delta_{z_n}(\mathrm{d}x_n^N) \prod_{i=1}^{N-1} p_n^y(\mathrm{d}x_n^i)$$

$$\times \prod_{\ell=2}^n \delta_{z_{\ell-1}}(\mathrm{d}x_{\ell-1}^N)\delta_N(\mathrm{d}a_{\ell-1}^N) \prod_{j=1}^{N-1}\sum_{k=1}^N \omega_{\ell-1}^k \delta_k(\mathrm{d}a_\ell^j) p_{\ell-1}^y(\mathrm{d}x_{\ell-1}^j|x_\ell^{a_\ell^j}) \quad \text{(B.4)}$$

$$\times \delta_{z_0}(\mathrm{d}x_0^N)\delta_N(\mathrm{d}a_1^N) \prod_{j=1}^{N-1}\sum_{k=1}^N \omega_0^k \delta_k(\mathrm{d}a_1^j) \underline{p}_0^y(\mathrm{d}\underline{x}_0^j|x_1^{a_1^j})\delta_y(\mathrm{d}\overline{x}_0^j).$$

In what follows $\mathbb{E}_{z_{0:n}}$ refers to expectation with respect to $\mathbf{P}^N(\cdot|z_{0:n})$. Finally, for $s \in [0:n-1]$ we let $\Omega_s^N$ denote the sum of the filtering weights at step $s$, i.e. $\Omega_s^N = \sum_{i=1}^N \widetilde{\omega}_s(\xi_{s+1}^i)$. We also write $\mathcal{Z}_0 = \int \mathsf{p}_0(x_0)\delta_y(\mathrm{d}\overline{x}_0)\mathrm{d}\underline{x}_0$ and for all $\ell \in [1:n]$, $\mathcal{Z}_\ell = \int \overline{q}_{\ell|0}(\overline{x}_\ell|y)\mathsf{p}_\ell(\mathrm{d}x_\ell)$.

The proof of Proposition 2.2 relies on two Lemmata stated below and proved in Appendix B.1; in Lemma B.1 we provide an expression for the Radon-Nikodym derivative $\mathrm{d}\phi_{0:n}^y/\mathrm{d}\Phi_{0:n}^y$ and in Lemma B.2 we explicit its leading term.

**Lemma B.1.** $\phi_{0:n}^y$ and $\Phi_{0:n}^N$ *are equivalent and we have that*

$$\Phi_{0:n}^N(\mathrm{d}z_{0:n}) = \mathbb{E}_{z_{0:n}}\left[\frac{N^n\,\mathcal{Z}_0/\mathcal{Z}_n}{\prod_{s=0}^{n-1}\Omega_s^N}\right]\phi_{0:n}^y(\mathrm{d}z_{0:n}). \tag{B.5}$$

**Lemma B.2.** *It holds that*

$$\frac{\mathcal{Z}_n}{\mathcal{Z}_0}\mathbb{E}_{z_{0:n}}\left[\prod_{s=0}^{n-1}N^{-1}\Omega_s^N\right] = \left(\frac{N-1}{N}\right)^n$$

$$+ \frac{(N-1)^{n-1}}{N^n}\sum_{s=1}^{n}\frac{\mathcal{Z}_s/\mathcal{Z}_0}{\overline{q}_{s|0}(\overline{z}_s|y)}\int p_{0|s}(x_0|z_s)\delta_y(\mathrm{d}\overline{x}_0)\mathrm{d}\underline{x}_0 + \frac{\mathrm{D}_{0:n}^y}{N^2}. \quad \text{(B.6)}$$

*where $\mathrm{D}_{0:n}^y$ is a positive constant.*

Before proceeding with the proof of Proposition 2.2, let us note that having $z \mapsto \widetilde{\omega}_\ell(z)$ bounded on $\mathbb{R}^{\mathrm{d}_x}$ for all $\ell \in [0:n-1]$ is sufficient to guarantee that $\mathrm{C}_{0:n}^y$ and $\mathrm{D}_{0:n}^y$ are finite since in this case it follows immediately that $\mathbb{E}_{z_{0:n}}\left[\prod_{s=0}^{n-1}N^{-1}\Omega_s^N\right]$ is bounded and so is the right hand side of equation B.6. This can be achieved with a slight modification of equation 2.5 and equation **??**. Indeed, consider instead the following recursion for $s \in [0:n]$ where $\delta > 0$,

$$\phi_n^y(x_n) \propto \left(\overline{q}_{n|0}(\overline{x}_n|y) + \delta\right)\mathrm{p}_n(x_n),$$

$$\phi_s^y(x_s) \propto \int \phi_{s+1}^y(x_{s+1})p_s(\mathrm{d}x_s|x_{s+1})\frac{\overline{q}_s(\overline{x}_s|y) + \delta}{\overline{q}_{s+1}(\overline{x}_{s+1}|y) + \delta}\mathrm{d}x_{s+1}.$$

Then we have that

$$\phi_0^y(\underline{x}_0) \propto \int \phi_1^y(x_1)\underline{p}_0(\underline{x}_0|x_1)\frac{\overline{p}_0(y|x_1)}{\overline{q}_{1|0}(\overline{x}_1|y) + \delta}\mathrm{d}x_1.$$

We can then use Algorithm 1 to produce a particle approximation of $\phi_0^y$ using the following transition and weight function,

$$p_s^{y,\delta}(x_s|x_{s+1}) = \frac{\gamma_s(y|x_{s+1})}{\gamma_s(y|x_{s+1}) + \delta}p_s^y(x_s|x_{s+1}) + \frac{\delta}{\gamma_s(y|x_{s+1}) + \delta}p_s(x_s|x_{s+1}),$$

$$\widetilde{\omega}_s(x_{s+1}) = \left(\gamma_s(y|x_{s+1}) + \delta\right)/\left(\overline{q}_{s+1|0}(\overline{x}_{s+1}|y) + \delta\right),$$

where $\gamma_s(y|x_{s+1}) = \int \overline{q}_{s|0}(\overline{x}_s|y)p_s(x_s|x_{s+1})\mathrm{d}x_s$ is available in closed form and $p_s^y$ is defined in equation 2.3. $\widetilde{\omega}_s$ is thus clearly bounded for all $s \in [0:n-1]$ and it is still possible to sample from $p_s^{y,\delta}$ since it is simply a mixture between the transition equation 2.3 and the "prior" transition.

*Proof of Proposition 2.2.* Consider the *forward* Markov kernel

$$\overrightarrow{\mathbf{B}}_{1:n}(z_0, \mathrm{d}z_{1:n}) = \frac{\mathrm{p}_{1:n}(\mathrm{d}z_{1:n})p_0(z_0|z_1)}{\int \mathrm{p}_{1:n}(\mathrm{d}\tilde{z}_{1:n})p_0(\tilde{z}_0|\tilde{z}_1)}, \quad \text{(B.7)}$$

which satisfies

$$\phi_{0:n}^y(\mathrm{d}z_{0:n}) = \phi_0^y(\mathrm{d}z_0)\overrightarrow{\mathbf{B}}_{1:n}(z_0, \mathrm{d}z_{1:n}).$$

By Lemma B.1 we have for any $C \in \mathcal{B}(\mathbb{R}^{\mathrm{d}_x})$ that

$$\Phi_0^N(C) = \int \Phi_{0:n}^N(\mathrm{d}z_{0:n})\mathbb{1}_C(z_0)$$

$$= \int \mathbb{1}_C(z_0)\mathbb{E}_{z_{0:n}}\left[\frac{N^n\mathcal{Z}_0/\mathcal{Z}_n}{\prod_{s=0}^{n-1}\Omega_s^N}\right]\phi_{0:n}^y(\mathrm{d}z_{0:n})$$

$$= \int \mathbb{1}_C(z_0)\int \overrightarrow{\mathbf{B}}_{1:n}(z_0, \mathrm{d}z_{1:n})\mathbb{E}_{z_{0:n}}\left[\frac{N^n\mathcal{Z}_0/\mathcal{Z}_n}{\prod_{s=0}^{n-1}\Omega_s^N}\right]\phi_0^y(\mathrm{d}z_0),$$

which shows that the Radon-Nikodym derivative $\mathrm{d}\Phi_0^N/\mathrm{d}\phi_0^y$ is,

$$\frac{\mathrm{d}\Phi_0^N}{\mathrm{d}\phi_0^y}(z_0) = \int \overrightarrow{\mathbf{B}}_{1:n}(z_0, \mathrm{d}z_{1:n}) \mathbb{E}_{z_{0:n}} \left[ \frac{N^n \mathcal{Z}_0/\mathcal{Z}_n}{\prod_{s=0}^{n-1} \Omega_s^N} \right].$$

Applying Jensen's inequality twice yields

$$\frac{\mathrm{d}\Phi_0^N}{\mathrm{d}\phi_0^y}(z_0) \geq \frac{N^n \mathcal{Z}_0/\mathcal{Z}_n}{\int \overrightarrow{\mathbf{B}}_{1:n}(z_0, \mathrm{d}z_{1:n}) \mathbb{E}_{z_{0:n}} \left[ \prod_{s=0}^{n-1} \Omega_s^N \right]},$$

and it then follows that

$$\mathsf{KL}(\phi_0^y \parallel \Phi_0^N) \leq \int \log \left( \frac{\mathcal{Z}_n}{\mathcal{Z}_0} \int \overrightarrow{\mathbf{B}}_{1:n}(z_0, \mathrm{d}z_{1:n}) \mathbb{E}_{z_{0:n}} \left[ \prod_{s=0}^{n-1} N^{-1} \Omega_s^N \right] \right) \phi_0^y(\mathrm{d}z_0).$$

Finally, using Lemma B.2 and the fact that $\log(1 + x) < x$ for $x > 0$ we get

$$\mathsf{KL}(\phi_0^y \parallel \Phi_0^N) \leq \frac{\mathsf{C}_{0:n}^y}{N-1} + \frac{\mathsf{D}_{0:n}^y}{N^2}$$

where

$$\mathsf{C}_{0:n}^y := \sum_{s=1}^n \int \frac{\mathcal{Z}_s/\mathcal{Z}_0}{\overline{q}_{s|0}(\overline{z}_s|y)} \left( p_{0|s}(x_0|z_s) \delta_y(\mathrm{d}\overline{x}_0) \mathrm{d}\underline{x}_0 \right) \phi_s^y(\mathrm{d}z_s),$$

and $\phi_s^y(z_s) \propto \mathsf{p}_s(z_s) \int p_{0|s}(z_0|z_s) \delta_y(\mathrm{d}\underline{z}_0) \mathrm{d}\overline{z}_0.$ □

### PROOF OF LEMMA B.1 AND LEMMA B.2

*Proof of Lemma B.1.* We have that

$$\Phi_{0:n}^N(\mathrm{d}z_{0:n})$$

$$= N^{-1} \int P_{0:n}^N(\mathrm{d}x_{0:n}^{1:N}, \mathrm{d}a_{1:n}^{1:N}) \sum_{k_{0:n} \in [1:N]^{n+1}} \delta_{y \frown \underline{x}_0^{k_0}}(\mathrm{d}z_0) \prod_{s=1}^n \mathbb{1}\{k_s = a_s^{k_{s-1}}\} \delta_{x_s^{k_s}}(\mathrm{d}z_s)$$

$$= N^{-1} \int \sum_{k_{0:n}} \sum_{a_{1:n}^{1:N}} \delta_{y \frown \underline{x}_0^{k_0}}(\mathrm{d}z_0) \prod_{s=1}^n \mathbb{1}\{k_s = a_s^{k_{s-1}}\} \delta_{x_s^{k_s}}(\mathrm{d}z_s)$$

$$\times \prod_{j=1}^N p_n^y(\mathrm{d}x_n^j) \left\{ \prod_{\ell=2}^n \prod_{i=1}^N \omega_{\ell-1}^{a_\ell^i} p_{\ell-1}^y(\mathrm{d}x_{\ell-1}^i | x_\ell^{a_\ell^i}) \right\} \prod_{r=1}^N \omega_0^{a_1^r} \underline{p}_{\ell-1}^y(\mathrm{d}\underline{x}_0^r | x_1^{a_1^r}) \delta_y(\overline{x}_0^r)$$

$$= N^{-1} \int \sum_{k_{0:n}} \sum_{a_{1:n}^{1:N}} p_n^y(\mathrm{d}x_n^{k_n}) \delta_{x_n^{k_n}}(\mathrm{d}z_n) \prod_{j \neq k_n} p_n^y(\mathrm{d}x_n^j) \prod_{\ell=2}^n \left\{ \prod_{i \neq k_{\ell-1}} \omega_{\ell-1}^{a_\ell^i} p_{\ell-1}^y(\mathrm{d}x_{\ell-1}^i | x_\ell^{a_\ell^i}) \right.$$

$$\times \mathbb{1}\{a_\ell^{k_{\ell-1}} = k_\ell\} \frac{\tilde{\omega}_{\ell-1}(x_\ell^{a_\ell^{k_{\ell-1}}})}{\Omega_{\ell-1}^N} p_{\ell-1}^y(\mathrm{d}x_{\ell-1}^{k_{\ell-1}} | x_\ell^{a_\ell^{k_{\ell-1}}}) \delta_{x_{\ell-1}^{k_{\ell-1}}}(\mathrm{d}z_{\ell-1}) \right\}$$

$$\times \left\{ \prod_{r \neq k_0} \omega_0^{a_1^r} \underline{p}_0^y(\mathrm{d}\underline{x}_0^r | x_1^{a_1^r}) \delta_y(\mathrm{d}\overline{x}_0^r) \right\} \mathbb{1}\{a_1^{k_0} = k_1\} \frac{\tilde{\omega}_0(x_1^{a_1^{k_0}})}{\Omega_0^N} p_0^y(\mathrm{d}\underline{x}_0^{k_0} | x_0^{a_1^{k_0}}) \delta_{y \frown \underline{x}_0^{k_0}}(\mathrm{d}z_0).$$

Then, using that for all $s \in [2:n]$

$$\widetilde{\omega}_{s-1}(x_s^{k_s}) p_{s-1}^y(\mathrm{d}x_{s-1}^{k_{s-1}}|x_s^{k_s}) = \frac{\overline{q}_{s-1|0}(\overline{x}_{s-1}^{k_{s-1}}|y)}{\overline{q}_{s|0}(\overline{x}_s^{k_s}|y)} p_s(\mathrm{d}x_{s-1}^{k_{s-1}}|x_s^{k_s}),$$

we recursively get that

$$p_n^y(\mathrm{d}x_n^{k_n}) \delta_{x_n^{k_n}}(\mathrm{d}z_n) \prod_{s=2}^{n} \mathbb{1}\{a_s^{k_{s-1}} = k_s\} \frac{\widetilde{\omega}_{s-1}(x_s^{a_s^{k_{s-1}}})}{\Omega_{s-1}^N} p_{s-1}^y(\mathrm{d}x_{s-1}^{k_{s-1}}|x_s^{a_s^{k_{s-1}}}) \delta_{x_{s-1}^{k_{s-1}}}(\mathrm{d}z_{s-1})$$

$$\times \mathbb{1}\{a_1^{k_0} = k_1\} \frac{\widetilde{\omega}_0(x_1^{a_1^{k_0}})}{\Omega_0^N} p_0^y(\mathrm{d}\underline{x}_0^{k_0}|x_1^{a_1^{k_0}}) \delta_{y\frown\underline{x}_0^{k_0}}(\mathrm{d}z_0)$$

$$= \frac{\overline{q}_{n|0}(z_n|y)\mathsf{p}_n(\mathrm{d}z_n)}{\mathcal{Z}_n} \delta_{z_n}(\mathrm{d}x_n^{k_n}) \prod_{s=2}^{n} \mathbb{1}\{a_s^{k_{s-1}} = k_s\} \frac{\overline{q}_{s-1|0}(\overline{z}_{s-1}|y)}{\Omega_{s-1}^N \overline{q}_{s|0}(\overline{z}_s|y)} p_{s-1}(\mathrm{d}z_{s-1}|z_s) \delta_{z_{s-1}}(\mathrm{d}x_{s-1}^{k_{s-1}})$$

$$\times \mathbb{1}\{a_1^{k_0} = k_1\} \frac{\overline{p}_0(y|z_1)}{\Omega_0^N \overline{q}_{1|0}(\overline{z}_1|y)} p_0(\mathrm{d}\underline{z}_0|z_1) \delta_y(\mathrm{d}\overline{z}_0) \delta_{z_0}(\mathrm{d}x_0^{k_0})$$

$$= \frac{\mathcal{Z}_0}{\mathcal{Z}_n} \phi_{0:n}^y(\mathrm{d}z_{0:n}) \delta_{z_n}(\mathrm{d}x_n^{k_n}) \prod_{s=1}^{n} \mathbb{1}\{a_s^{k_{s-1}} = k_s\} \frac{1}{\Omega_{s-1}^N} \delta_{z_{s-1}}(\mathrm{d}x_{s-1}^{k_{s-1}}).$$

Thus, we obtain

$$\Phi_{0:n}^N(\mathrm{d}z_{0:n}) = N^{-1} \int \sum_{k_{0:n}} \sum_{a_{1:n}^{1:N}} \phi_{0:n}^y(\mathrm{d}z_{0:n}) \frac{\mathcal{Z}_0/\mathcal{Z}_n}{\prod_{s=0}^{n-1} \Omega_s^N} \delta_{z_n}(\mathrm{d}x_n^{k_n}) \prod_{j \neq k_n} p_n^y(\mathrm{d}x_n^j)$$

$$\times \prod_{\ell=2}^{n} \mathbb{1}\{a_\ell^{k_{\ell-1}} = k_\ell\} \delta_{z_{\ell-1}}(\mathrm{d}x_{\ell-1}^{k_{\ell-1}}) \prod_{i \neq k_{\ell-1}} \omega_{\ell-1}^{a_\ell^i} p_{\ell-1}^y(\mathrm{d}x_{\ell-1}^i|x_\ell^{a_\ell^i})$$

$$\times \mathbb{1}\{a_1^{k_0} = k_1\} \delta_{z_0}(\mathrm{d}x_0^{k_0}) \prod_{i \neq k_0} \omega_0^{a_1^i} \underline{p}_0(\underline{x}_0^i|x_1^{a_1^i}) \delta_y(\mathrm{d}\overline{x}_0^i)$$

$$= N^{-1} \sum_{k_{0:n}} \phi_{0:n}^y(\mathrm{d}z_{0:n}) \mathbb{E}_{z_{0:n}}^{k_{0:n}} \left[ \frac{\mathcal{Z}_0/\mathcal{Z}_n}{\prod_{s=0}^{n-1} \Omega_s^N} \right],$$

where for all $k_{0:n} \in [1:N]^{n+1}$ $\mathbb{E}_{z_{0:n}}^{k_{0:n}}$ denotes the expectation under the Markov kernel

$$\mathbf{P}_{k_{0:n}}^N\left(\mathrm{d}(x_{0:n}^{1:N}, a_{1:n}^{1:N})\big|z_{0:n}\right) = \delta_{z_n}(\mathrm{d}x_n^{k_n}) \prod_{i \neq k_n} p_n^y(\mathrm{d}x_n^i)$$

$$\times \prod_{\ell=2}^{n} \delta_{z_{\ell-1}}(\mathrm{d}x_{\ell-1}^{k_{\ell-1}}) \delta_{k_\ell}(\mathrm{d}a_\ell^{k_{\ell-1}}) \prod_{j \neq k_{\ell-1}} \sum_{k=1}^{N} \omega_{\ell-1}^k \delta_k(\mathrm{d}a_\ell^j) p_{\ell-1}^y(\mathrm{d}x_{\ell-1}^j|x_\ell^{a_\ell^j})$$

$$\times \delta_{z_0}(\mathrm{d}x_0^{k_0}) \delta_{k_1}(\mathrm{d}a_1^{k_0}) \prod_{j \neq k_0} \sum_{k=1}^{N} \omega_0^k \delta_k(\mathrm{d}a_1^j) \underline{p}_0^y(\mathrm{d}\underline{x}_0^j|x_1^{a_1^j}) \delta_y(\mathrm{d}\overline{x}_0).$$

Note however that for all $(k_{0:n}, \ell_{0:n}) \in ([1:N]^{n+1})^2$,

$$\mathbb{E}_{z_{0:n}}^{k_{0:n}} \left[ \frac{1}{\prod_{s=0}^{n-1} \Omega_s^N} \right] = \mathbb{E}_{z_{0:n}}^{\ell_{0:n}} \left[ \frac{1}{\prod_{s=0}^{n-1} \Omega_s^N} \right]$$

and thus it follows that

$$\Phi_{0:n}^N(\mathrm{d}z_{0:n}) = \mathbb{E}_{z_{0:n}} \left[ \frac{N^n \mathcal{Z}_0/\mathcal{Z}_n}{\prod_{s=0}^{n-1} \Omega_s^N} \right] \phi_{0:n}^y(\mathrm{d}z_{0:n}) . \tag{B.8}$$

$\square$

Denote by $\{\mathcal{F}_s\}_{s=0}^n$ the filtration generated by a conditional particle cloud sampled from the kernel $\mathbf{P}^N$ equation B.4, i.e. for all $\ell \in [0:n-1]$

$$\mathcal{F}_s = \sigma\left(\xi_{s:n}^{1:N}, I_{s+1:n}^{1:N}\right) .$$

and $\mathcal{F}_n = \sigma\left(\xi_n^{1:N}\right)$. Define for all bounded $f$ and $\ell \in [0:n-1]$

$$\gamma_{\ell:n}^N(f) = \left\{ \prod_{s=\ell+1}^{n-1} N^{-1}\Omega_s^N \right\} N^{-1} \sum_{k=1}^N \widetilde{\omega}_\ell(\xi_{\ell+1}^k) f(\xi_{\ell+1}^k) , \tag{B.9}$$

with the convention $\gamma_{\ell:n}^N(f) = 1$ if $\ell \geq n$. Define also the transition Kernel

$$Q_{\ell-1|\ell+1}^y : \mathbb{R}^{\mathrm{d}_x} \times \mathcal{B}(\mathbb{R}^{\mathrm{d}_x}) \ni (x_{\ell+1}, A) \mapsto \int \mathbb{1}_A(x_\ell)\widetilde{\omega}_{\ell-1}(x_\ell)p_\ell^y(\mathrm{d}x_\ell|x_{\ell+1}) . \tag{B.10}$$

Using eqs. (2.3) and (2.4), it is easily seen that for all $\ell \in [0:n-1]$,

$$\widetilde{\omega}_\ell(x_{\ell+1})Q_{\ell-1|\ell+1}^y(f)(x_{\ell+1}) = \frac{1}{\overline{q}_{\ell+1|0}(\overline{x}_{\ell+1}|y)} \int \overline{q}_{\ell|0}(\overline{x}_s|y)\widetilde{\omega}_{\ell-1}(x_\ell)f(x_\ell)p_\ell(\mathrm{d}x_\ell|x_{\ell+1}) . \tag{B.11}$$

Define $\mathbf{1} : x \in \mathbb{R}^{\mathrm{d}_x} \mapsto 1$. We may thus write that $\gamma_{\ell:n}^N(f) = N^{-1}\gamma_{\ell+1:n}^N(\mathbf{1}) \sum_{k=1}^N \widetilde{\omega}_\ell(\xi_{\ell+1}^k) f(\xi_{\ell+1}^k)$.

**Lemma B.3.** *For all $\ell \in [0:n-1]$ it holds that*

$$\mathbb{E}_{z_{0:n}} \left[\gamma_{\ell-1:n}^N(f)\right] = \frac{N-1}{N}\mathbb{E}_{z_{0:n}} \left[\gamma_{\ell:n}^N \left(Q_{\ell-1|\ell+1}^y(f)\right)\right] + \frac{1}{N}\mathbb{E}_{z_{0:n}} \left[\gamma_{\ell:n}^N(\mathbf{1})\right] \widetilde{\omega}_{\ell-1}(z_\ell)f(z_\ell) .$$

*Proof.* By the tower property and the fact that $\gamma_{\ell:n}^N(f)$ is $\mathcal{F}_{\ell+1}$-measurable, we have that

$$\mathbb{E}_{z_{0:n}} \left[\gamma_{\ell-1:n}^N(f)\right] = \mathbb{E}_{z_{0:n}} \left[N^{-1}\gamma_{\ell+1:n}^N(\mathbf{1})\Omega_\ell^N \mathbb{E}_{z_{0:n}} \left[N^{-1} \sum_{k=1}^N \widetilde{\omega}_{\ell-1}(\xi_\ell^k)f(\xi_\ell^k)\Big|\mathcal{F}_{\ell+1}\right]\right] .$$

Note that for all $\ell \in [0:n-1]$, $(\xi_\ell^1, \ldots, \xi_\ell^{N-1})$ are identically distributed conditionally on $\mathcal{F}_{\ell+1}$ and

$$\mathbb{E}_{z_{0:n}} \left[\widetilde{\omega}_{\ell-1}(\xi_\ell^j)f(\xi_\ell^j)\Big|\mathcal{F}_{\ell+1}\right] = \frac{1}{\Omega_\ell^N} \sum_{k=1}^N \widetilde{\omega}_\ell(\xi_{\ell+1}^k) \int \widetilde{\omega}_{\ell-1}(x_\ell)f(x_\ell)p_\ell^y(\mathrm{d}x_\ell|\xi_{\ell+1}^k) ,$$

leading to

$$\mathbb{E}_{z_{0:n}} \left[N^{-1} \sum_{k=1}^N \widetilde{\omega}_{\ell-1}(\xi_\ell^k)f(\xi_\ell^k)\Big|\mathcal{F}_{\ell+1}\right]$$
$$= \frac{N-1}{N\Omega_\ell^N} \sum_{k=1}^N \widetilde{\omega}_\ell(\xi_{\ell+1}^k) \int \widetilde{\omega}_{\ell-1}(x_\ell)f(x_\ell)p_\ell^y(\mathrm{d}x_\ell|\xi_{\ell+1}^k) + \frac{1}{N}\widetilde{\omega}_{\ell-1}(z_\ell)f(z_\ell) ,$$

and the desired recursion follows. $\square$

*Proof of Lemma B.2.* We proceed by induction and show for all $\ell \in [0:n-2]$

$$
\begin{aligned}
&\mathbb{E}_{z_{0:n}}\left[\gamma^N_{\ell:n}(f)\right] \\
&= \left(\frac{N-1}{N}\right)^{n-\ell} \frac{\int \mathsf{p}_{\ell+1}(\mathrm{d}x_{\ell+1})\overline{q}_{\ell+1|0}(\overline{x}_{\ell+1}|y)\widetilde{\omega}_\ell(x_{\ell+1})f(x_{\ell+1})}{\mathcal{Z}_n} \\
&\quad + \frac{(N-1)^{n-\ell-1}}{N^{n-\ell}}\Big[(\mathcal{Z}_{\ell+1}/\mathcal{Z}_n)f(z_{\ell+1})\widetilde{\omega}_\ell(z_{\ell+1}) \\
&\quad + \sum_{s=\ell+2}^n \frac{\mathcal{Z}_s/\mathcal{Z}_n}{\overline{q}_{s|0}(\overline{z}_s|y)} \int \widetilde{\omega}_\ell(x_{\ell+1})\overline{q}_{\ell+1|0}(\overline{x}_{\ell+1}|y)f(x_{\ell+1})p_{\ell+1|s}(\mathrm{d}x_{\ell+1}|z_s)\Big] + \frac{\mathsf{D}^y_{\ell:n}}{N^2}.
\end{aligned}
\tag{B.12}
$$

where $f$ is a bounded function and $\mathsf{D}^y_{\ell:n}$ is a a positive constant. The desired result in Lemma B.2 then follows by taking $\ell = 0$ and $f = \mathbf{1}$.

Assume that equation B.12 holds at step $\ell$. To show that it holds at step $\ell-1$ we use Lemma B.3 and we compute $\mathbb{E}_{z_{0:n}}\left[\gamma^N_{\ell:n}\left(Q^y_{\ell-1|\ell+1}(f)\right)\right]$ and $\mathbb{E}_{z_{0:n}}\left[\gamma^N_{\ell:n}(\mathbf{1})\right]\widetilde{\omega}_{\ell-1}(z_\ell)f(z_\ell)$.

Using the following identities which follow from equation B.11

$$
\int \overline{q}_{\ell+1|0}(\overline{x}_{\ell+1}|y)\widetilde{\omega}_\ell(x_{\ell+1})Q^y_{\ell-1|\ell+1}(f)(x_{\ell+1})\mathsf{p}_{\ell+1}(\mathrm{d}x_{\ell+1})
$$
$$
= \int \overline{q}_{\ell|0}(\overline{x}_\ell|y)\widetilde{\omega}_{\ell-1}(x_\ell)f(x_\ell)\mathsf{p}_\ell(\mathrm{d}x_\ell),
$$

and

$$
\int \widetilde{\omega}_\ell(x_{\ell+1})\overline{q}_{\ell+1|0}(\overline{x}_{\ell+1}|y)Q^y_{\ell-1|\ell+1}(f)(x_{\ell+1})p_{\ell+1|s}(\mathrm{d}x_{\ell+1}|x_s)
$$
$$
= \int \widetilde{\omega}_{\ell-1}(x_\ell)\overline{q}_{\ell|0}(\overline{x}_\ell|y)f(x_\ell)p_{\ell|s}(\mathrm{d}x_\ell|x_s),
$$

we get by equation B.12 that

$$
\begin{aligned}
&\frac{N-1}{N}\mathbb{E}_{z_{0:n}}\left[\gamma^N_{\ell:n}\left(Q^y_{\ell-1|\ell+1}(f)\right)\right] \\
&= \left(\frac{N-1}{N}\right)^{n-\ell+1} \frac{\int \overline{q}_{\ell|0}(\overline{x}_\ell|y)\widetilde{\omega}_{\ell-1}(x_\ell)f(x_\ell)\mathsf{p}_\ell(\mathrm{d}x_\ell)}{\mathcal{Z}_n} \\
&\quad + \frac{(N-1)^{n-\ell}}{N^{n-\ell+1}}\Big[\frac{\mathcal{Z}_{\ell+1}/\mathcal{Z}_n}{\overline{q}_{\ell+1|0}(\overline{z}_{\ell+1}|y)} \int \overline{q}_{\ell|0}(\overline{x}_\ell|y)\widetilde{\omega}_{\ell-1}(x_\ell)f(x_\ell)p_\ell(\mathrm{d}x_\ell|z_{\ell+1}) \\
&\quad + \sum_{s=\ell+2}^n \frac{\mathcal{Z}_s/\mathcal{Z}_n}{\overline{q}_{s|0}(\overline{z}_s|y)} \int \widetilde{\omega}_{\ell-1}(x_\ell)\overline{q}_{\ell|0}(\overline{x}_\ell|y)f(x_\ell)p_{\ell|s}(\mathrm{d}x_\ell|z_s)\Big] + \frac{\mathsf{D}^y_{\ell:n}}{N^2} \\
&= \left(\frac{N-1}{N}\right)^{n-\ell+1} \frac{\int \overline{q}_{\ell|0}(\overline{x}_\ell|y)\widetilde{\omega}_{\ell-1}(x_\ell)f(x_\ell)\mathsf{p}_\ell(\mathrm{d}x_\ell)}{\mathcal{Z}_n} \\
&\quad + \frac{(N-1)^{n-\ell}}{N^{n-\ell+1}} \sum_{s=\ell+1}^n \frac{\mathcal{Z}_s/\mathcal{Z}_n}{\overline{q}_{s|0}(\overline{z}_s|y)} \int \widetilde{\omega}_{\ell-1}(x_\ell)\overline{q}_{\ell|0}(\overline{x}_s|y)f(x_\ell)p_{\ell|s}(\mathrm{d}x_\ell|z_s) + \frac{\mathsf{D}^y_{\ell:n}}{N^2}.
\end{aligned}
\tag{B.13}
$$

The induction step is finished by using again equation B.12 and noting that

$$\frac{1}{N}\mathbb{E}_{z_{0:n}}\left[\gamma_{\ell:n}^N(\mathbf{1})\right]\widetilde{\omega}_{\ell-1}(z_\ell)f(z_\ell) = \frac{(N-1)^{n-\ell}}{N^{n-\ell+1}}\left(\mathcal{Z}_\ell/\mathcal{Z}_n\right)\widetilde{\omega}_{\ell-1}(z_\ell)f(z_\ell) + \frac{\widetilde{\mathsf{D}}_{\ell:n}^y}{N^2}\,.$$

and then setting $\mathsf{D}_{\ell-1:n}^y = \mathsf{D}_{\ell:n}^y + \widetilde{\mathsf{D}}_{\ell:n}^y$.

It remains to compute the initial value at $\ell = n-2$. Note that

$$\mathbb{E}_{z_{0:n}}\left[\gamma_{n-1:n}^N(f)\right] = \frac{N-1}{N}\int p_n^y(\mathrm{d}x_n)\widetilde{\omega}_{n-1}(x_n)f(x_n) + \frac{1}{N}\widetilde{\omega}_{n-1}(z_n)f(z_n) \tag{B.14}$$

and thus by Lemma B.3 and similarly to the previous computations

$$\begin{aligned}
&\mathbb{E}_{z_{0:n}}\left[\gamma_{n-2:n}^N(f)\right]\\
&= \left(\frac{N-1}{N}\right)^2\int p_n^y(\mathrm{d}x_n)\widetilde{\omega}_{n-1}(x_n)Q_{n-2|n}^y(f)(x_n) + \frac{N-1}{N^2}\Big[\widetilde{\omega}_{n-1}(z_n)Q_{n-2|n}^y(f)(z_n)\\
&\quad + \widetilde{\omega}_{n-2}(z_{n-1})f(z_{n-1})\int p_n^y(\mathrm{d}x_n)\widetilde{\omega}_{n-1|n}(x_n)\Big] + \frac{\mathsf{D}_{n-2fa:n}^y}{N^2}\\
&= \left(\frac{N-1}{N}\right)^2\frac{\int \bar{q}_{n-1|0}(x_{n-1}|y)\widetilde{\omega}_{n-2}(x_{n-1})\mathsf{p}_{n-1}(\mathrm{d}x_{n-1})}{\mathcal{Z}_n}\\
&\quad + \frac{N-1}{N^2}\Big[(\mathcal{Z}_{n-1}/\mathcal{Z}_n)\widetilde{\omega}_{n-2}(z_{n-1})f(z_{n-1})\\
&\quad + \frac{1}{\bar{q}_{n|0}(\overline{x}_n|y)}\int \bar{q}_{n-1|0}(\overline{x}_{n-1}|y)\widetilde{\omega}_{n-2}(x_{n-1})f(x_{n-1})p_{n-1}(\mathrm{d}x_{n-1}|z_n)\Big] + \frac{\mathsf{D}_{n-2:n}^y}{N^2}\,.
\end{aligned}$$

$\square$

## B.2 PROOF OF PROPOSITION 2.3 AND LEMMA B.4

In this section and only in this section we make the following assumption

(A2) For all $s \in [0:n-1]$, $\mathfrak{p}_s(x_s)q_{s+1}(x_{s+1}|x_s) = \mathfrak{p}_{s+1}(x_{s+1})\lambda_s(x_s|x_{s+1})$.

We also consider $\sigma_\delta = 0$. In what follows we let $\tau_{\mathsf{d_y}+1} = n$ and we write $\tau_{1:\mathsf{d_y}} = \{\tau_1,\ldots,\tau_{\mathsf{d_y}}\}$ and $\overline{\tau_{1:\mathsf{d_y}}} = [1:n]\setminus\tau_{1:t}$. Define the measure

$$\Gamma_{0:n}^{\mathbf{y}}(\mathrm{d}\mathbf{x}_{0:n}) = \mathfrak{p}_n(\mathrm{d}\mathbf{x}_n)\prod_{s\in\overline{\tau_{1:\mathsf{d_y}}}}\lambda_s(\mathrm{d}\mathbf{x}_s|\mathbf{x}_{s+1})\prod_{i=1}^{\mathsf{d_y}}\lambda_{\tau_i}(\mathbf{x}_{\tau_i}|\mathbf{x}_{\tau_i+1})\mathrm{d}\mathbf{x}_{\tau_i}^{\setminus i}\delta_{\mathbf{y}[i]}(\mathrm{d}\mathbf{x}_{\tau_i}[i])\,. \tag{B.15}$$

Under (A2) it has the following alternative *forward* expression,

$$\Gamma_{0:n}^{\mathbf{y}}(\mathrm{d}\mathbf{x}_{0:n}) = \mathfrak{p}_0(\mathrm{d}\mathbf{x}_0)\prod_{s\in\overline{\tau_{1:\mathsf{d_y}}}}q_{s+1}(\mathrm{d}\mathbf{x}_{s+1}|\mathbf{x}_s)\prod_{i=1}^{\mathsf{d_y}}q_{\tau_i}(\mathbf{x}_{\tau_i}|\mathbf{x}_{\tau_i-1})\mathrm{d}\mathbf{x}_{\tau_i}^{\setminus i}\delta_{\mathbf{y}[i]}(\mathrm{d}\mathbf{x}_{\tau_i}[i])\,. \tag{B.16}$$

Since the forward kernels decompose over the dimensions of the states, i.e.

$$q_{s+1}(\mathbf{x}_{s+1}|\mathbf{x}_s) = \prod_{\ell=1}^{\mathsf{d_x}}q_{s+1}^\ell(\mathbf{x}_{s+1}[\ell]|\mathbf{x}_s[\ell])$$

where $q_{s+1}^\ell(\mathbf{x}_{s+1}[\ell]|\mathbf{x}_s[\ell]) = \mathcal{N}(\mathbf{x}_{s+1}[\ell]; (\alpha_{s+1}/\alpha_s)^{1/2}\mathbf{x}_s[\ell], 1 - (\alpha_{s+1}/\alpha_s))$, we can write

$$\Gamma_{0:n}^{\mathbf{y}}(\mathbf{x}_{0:n}) = \mathfrak{p}_0(\mathbf{x}_0) \prod_{\ell=1}^{\mathsf{d}_x} \Gamma_{1:n|0,\ell}^{\mathbf{y}}(\mathbf{x}_1[\ell], \ldots, \mathbf{x}_n[\ell]|\mathbf{x}_0[\ell]), \tag{B.17}$$

where for $\ell \in [1 : \mathsf{d}_{\mathbf{y}}]$

$$\Gamma_{1:n|0,\ell}^{\mathbf{y}}(\mathbf{x}_1[\ell], \ldots, \mathbf{x}_n[\ell]|\mathbf{x}_0[\ell]) = q_{\tau_\ell}^\ell(\mathbf{y}[\ell]|\mathbf{x}_{\tau_\ell-1}[\ell]) \prod_{s \neq \tau_\ell} q_s^\ell(\mathrm{d}\mathbf{x}_s[\ell]|\mathbf{x}_{s-1}[\ell]), \tag{B.18}$$

and for $\ell \in [\mathsf{d}_{\mathbf{y}} + 1 : \mathsf{d}_x]$,

$$\Gamma_{1:n|0,\ell}^{\mathbf{y}}(\mathbf{x}_1[\ell], \ldots, \mathbf{x}_n[\ell]|\mathbf{x}_0[\ell]) = \prod_{s=0}^{n-1} q_{s+1}^\ell(\mathbf{x}_{s+1}[\ell]|\mathbf{x}_s[\ell]). \tag{B.19}$$

With these quantities in hand we can now prove Proposition 2.3.

*Proof of Proposition 2.3.* Note that for $\ell \in [1 : \mathsf{d}_{\mathbf{y}}]$,

$$\begin{aligned}
\mathcal{N}(\mathbf{y}[\ell]; \alpha_{\tau_\ell}\mathbf{x}_0[\ell], 1 - \alpha_{\tau_\ell}) = q_{\tau_\ell|0}^\ell(\mathbf{y}[\ell]|\mathbf{x}_0[\ell]) &= \int q_{\tau_\ell}^\ell(\mathbf{y}[\ell]|\mathbf{x}_{\tau_\ell-1}[\ell]) \prod_{s \neq \tau_\ell} q_s^\ell(\mathrm{d}\mathbf{x}_s[\ell]|\mathbf{x}_{s-1}[\ell]) \\
&= \int \Gamma_{1:n|0,\ell}^{\mathbf{y}}(\mathrm{d}(\mathbf{x}_1[\ell], \ldots, \mathbf{x}_n[\ell])|\mathbf{x}_0[\ell])
\end{aligned}$$

and thus by equation **??** we have that

$$\begin{aligned}
\mathfrak{p}_0(\mathbf{x}_0)g_0^y(\mathbf{x}_0) &\propto \mathfrak{p}_0(\mathbf{x}_0) \prod_{\ell=1}^{\mathsf{d}_{\mathbf{y}}} \mathcal{N}(\mathbf{y}[\ell]; \alpha_{\tau_\ell}\mathbf{x}_0[\ell], 1 - \alpha_{\tau_\ell}) \\
&= \mathfrak{p}_0(\mathbf{x}_0) \prod_{\ell=1}^{\mathsf{d}_{\mathbf{y}}} \int \Gamma_{1:n|0,\ell}^{\mathbf{y}}(\mathrm{d}(\mathbf{x}_1[\ell], \ldots, \mathbf{x}_n[\ell])|\mathbf{x}_0[\ell]) \\
&= \mathfrak{p}_0(\mathbf{x}_0) \prod_{\ell=1}^{\mathsf{d}_x} \int \Gamma_{1:n|0,\ell}^{\mathbf{y}}(\mathrm{d}(\mathbf{x}_1[\ell], \ldots, \mathbf{x}_n[\ell])|\mathbf{x}_0[\ell]).
\end{aligned}$$

By equation B.16 it follows that

$$\phi_0^{\mathbf{y}}(\mathbf{x}_0) = \frac{1}{\int \Gamma_{0:n}^{\mathbf{y}}(\tilde{\mathbf{x}}_{0:n})\mathrm{d}\tilde{\mathbf{x}}_{0:n}} \int \Gamma_{0:n}^{\mathbf{y}}(\mathbf{x}_{0:n})\mathrm{d}\mathbf{x}_{1:n},$$

and hence by equation B.16 and equation B.15 we get

$$\phi_0^{\mathbf{y}}(\mathbf{x}_0) \propto \int \mathfrak{p}_{\tau_{\mathsf{d}_{\mathbf{y}}}}(\mathbf{x}_{\tau_{\mathsf{d}_{\mathbf{y}}}})\delta_{\mathbf{y}[\mathsf{d}_{\mathbf{y}}]}(\mathrm{d}\mathbf{x}_{\tau_{\mathsf{d}_{\mathbf{y}}}}[\mathsf{d}_{\mathbf{y}}])\mathrm{d}\mathbf{x}_{\tau_{\mathsf{d}_{\mathbf{y}}}}^{\backslash\mathsf{d}_{\mathbf{y}}} \left\{ \prod_{i=1}^{\mathsf{d}_{\mathbf{y}}-1} \lambda_{\tau_i|\tau_{i+1}}(\mathbf{x}_{\tau_i}|\mathbf{x}_{\tau_{i+1}})\delta_{\mathbf{y}[i]}(\mathrm{d}\mathbf{x}_{\tau_i}[i])\mathrm{d}\mathbf{x}_{\tau_i}^{\backslash i} \right\} \lambda_{0|\tau_1}(\mathbf{x}_0|\mathbf{x}_{\tau_1}).$$

This completes the proof. $\square$

Let $\gamma_{0,s}^{\mathbf{y}}$ denote the joint time $0$ and $s$ marginal of the measure equation B.15, i.e.

$$\gamma_{0,s}^{\mathbf{y}}(\mathbf{x}_0, \mathbf{x}_s) = \int \Gamma_{0:n}^{\mathbf{y}}(\mathbf{x}_{0:n})\mathrm{d}\mathbf{x}_{1:s-1}\mathrm{d}\mathbf{x}_{s+1:n} \tag{B.20}$$

We now prove the following result.

**Lemma B.4.** *Assume (A2) and let $\tau_0 := 0$, $\tau_{\mathsf{d_y}+1} := n$. For all $k \in [1 : \mathsf{d_y}]$,*

*(i) If $s \in [\tau_k + 1 : \tau_{k+1}]$,*

$$\gamma_{0,s}^{\mathbf{y}}(\mathbf{x}_0, \mathbf{x}_s) =$$

$$\int \gamma_{0,s+1}^{\mathbf{y}}(\mathbf{x}_0, \mathbf{x}_{s+1})\underline{q}_{s|s+1,0}^{\sigma}(\underline{x}_s|\underline{x}_{s+1}, \underline{x}_0)g_s^{\mathbf{y}}(\overline{x}_s) \prod_{\ell=k+1}^{\mathsf{d_y}} q_{s|s+1,0}^{\sigma,\ell}(\mathbf{x}_s[\ell]|\mathbf{x}_{s+1}[\ell], \mathbf{x}_0[\ell]) \mathrm{d}\mathbf{x}_{s+1} \,.$$

*(ii) If $s = \tau_k$,*

$$\gamma_{0,s}^{\mathbf{y}}(\mathbf{x}_0, \mathbf{x}_s) = \int \gamma_{0,s+1}^{\mathbf{y}}(\mathbf{x}_0, \mathbf{x}_{s+1})\underline{q}_{s|s+1,0}^{\sigma}(\underline{x}_s|\underline{x}_{s+1}, \underline{x}_0)$$

$$\times \prod_{i=1}^{k-1} g_{s,i}^{\mathbf{y}}(\overline{x}_s[i]) \prod_{\ell=k+1}^{\mathsf{d_y}} q_{s|s+1,0}^{\sigma,\ell}(\mathbf{x}_s[\ell]|\mathbf{x}_{s+1}[\ell], \mathbf{x}_0[\ell]) \mathrm{d}\mathbf{x}_{s+1} \,.$$

*Proof of Lemma B.4.* Let $k \in [1 : \mathsf{d_y}]$ and assume that $s \in [\tau_k + 1 : \tau_{k+1} - 2]$. By (A2), equation B.16, equation B.18 and equation B.19 we have that

$$\gamma_{0,s}^{\mathbf{y}}(\mathbf{x}_0, \mathbf{x}_s) = \mathfrak{p}_0(\mathbf{x}_0)\underline{q}_{s|0}(\underline{x}_s|\underline{x}_0) \prod_{i=1}^{k} q_{\tau_i|0}^{i}(\mathbf{y}[i]|\mathbf{x}_0[i])q_{s|\tau_i}^{i}(\mathbf{x}_s[i]|\mathbf{y}[i])$$

$$\times \prod_{\ell=k+1}^{\mathsf{d_y}} q_{s|0}^{\ell}(\mathbf{x}_s[\ell]|\mathbf{x}_0[\ell])q_{\tau_\ell|s}^{\ell}(\mathbf{y}[\ell]|\mathbf{x}_s[\ell]) \,,$$

and thus, using the following identity valid for $\ell \in [k + 1 : \mathsf{d_y}]$

$$q_{s|0}^{\ell}(\mathbf{x}_s[\ell]|\mathbf{x}_0[\ell])q_{\tau_\ell|s}^{\ell}(\mathbf{y}[\ell]|\mathbf{x}_s[\ell])$$

$$= q_{s|0}^{\ell}(\mathbf{x}_s[\ell]|\mathbf{x}_0[\ell]) \int q_{\tau_\ell|s+1}^{\ell}(\mathbf{y}[\ell]|\mathbf{x}_{s+1}[\ell])q_{s+1}^{\ell}(\mathbf{x}_{s+1}[\ell]|\mathbf{x}_s[\ell]) \mathrm{d}\mathbf{x}_{s+1}[\ell]$$

$$= \int q_{s|s+1,0}^{\sigma,\ell}(\mathbf{x}_s[\ell]|\mathbf{x}_{s+1}[\ell], \mathbf{x}_0[\ell])q_{\tau_\ell|s+1}^{\ell}(\mathbf{y}[\ell]|\mathbf{x}_{s+1}[\ell])q_{s+1|0}^{\ell}(\mathbf{x}_{s+1}[\ell]|\mathbf{x}_0[\ell]) \mathrm{d}\mathbf{x}_{s+1}[\ell] \,,$$

and that $\underline{q}_{s|0}(\underline{x}_s|\underline{x}_0)\underline{q}_{s+1}(\underline{x}_{s+1}|\underline{x}_s) = \underline{q}^{\sigma}_{s|s+1,0}(\underline{x}_s|\underline{x}_{s+1},\underline{x}_0)\underline{q}_{s+1|0}(\underline{x}_{s+1}|\underline{x}_0)$ we get that

$$
\begin{aligned}
\gamma^{\mathbf{y}}_{0,s}&(\mathbf{x}_0,\mathbf{x}_s)\\
&= \int \mathfrak{p}_0(\mathbf{x}_0)\underline{q}_{s|0}(\underline{x}_s|\underline{x}_0)\underline{q}_{s+1}(\mathrm{d}\underline{x}_{s+1}|\underline{x}_s)\\
&\qquad\times \prod_{i=1}^{k} q^i_{\tau_i|0}(\mathbf{y}[i]|\mathbf{x}_0[i])q^i_{s|\tau_i}(\mathbf{x}_s[i]|\mathbf{y}[i])q^i_{s+1|\tau_i}(\mathrm{d}\mathbf{x}_{s+1}[i]|\mathbf{y}[i])\\
&\qquad\times \prod_{\ell=k+1}^{\mathbf{d_y}} q^{\sigma,\ell}_{s|s+1,0}(\mathbf{x}_s[\ell]|\mathbf{x}_{s+1}[\ell],\mathbf{x}_0[\ell])q^\ell_{\tau_\ell|s+1}(\mathbf{y}[\ell]|\mathbf{x}_{s+1}[\ell])q^\ell_{s+1|0}(\mathbf{x}_{s+1}[\ell]|\mathbf{x}_0[\ell])\mathrm{d}\mathbf{x}_{s+1}[\ell]\\
&= \int \gamma^{\mathbf{y}}_{0,s+1}(\mathbf{x}_0,\mathbf{x}_{s+1})\underline{q}^{\sigma}_{s|s+1,0}(\underline{x}_s|\underline{x}_{s+1},\underline{x}_0)g^{\mathbf{y}}_s(\overline{x}_s) \prod_{\ell=k+1}^{\mathbf{d_y}} q^{\sigma,\ell}_{s|s+1,0}(\mathbf{x}_s[\ell]|\mathbf{x}_{s+1}[\ell],\mathbf{x}_0[\ell])\mathrm{d}\mathbf{x}_{s+1}\,.
\end{aligned}
$$

If $s = \tau_{k+1}$ then

$$
\begin{aligned}
\gamma^{\mathbf{y}}_{0,s}(\mathbf{x}_0,\mathbf{x}_s) &= \mathfrak{p}_0(\mathbf{x}_0)\underline{q}_{s|0}(\underline{x}_s|\underline{x}_0) \prod_{i=1}^{k} q^i_{\tau_i|0}(\mathbf{y}[i]|\mathbf{x}_0[i])q^i_{s|\tau_i}(\mathbf{x}_s[i]|\mathbf{y}[i])\\
&\qquad\times q^{k+1}_{\tau_{k+1}|0}(\mathbf{y}[k+1]|\mathbf{x}_0[k+1]) \prod_{\ell=k+2}^{\mathbf{d_y}} q^\ell_{s|0}(\mathbf{x}_s[\ell]|\mathbf{x}_0[\ell])q^\ell_{\tau_\ell|s}(\mathbf{y}[\ell]|\mathbf{x}_s[\ell])\,,
\end{aligned}
\tag{B.21}
$$

and similarly to the previous case we get

$$
\begin{aligned}
\gamma_{0,s}&(\mathbf{x}_0,\mathbf{x}_s)\\
&= \int \gamma^{\mathbf{y}}_{0,s+1}(\mathbf{x}_0,\mathbf{x}_{s+1})\underline{q}^{\sigma}_{s|s+1,0}(\underline{x}_s|\underline{x}_{s+1},\underline{x}_0)g^{\mathbf{y}}_s(\overline{x}_s) \prod_{\ell=k+2}^{\mathbf{d_y}} q^{\sigma,\ell}_{s|s+1,0}(\mathbf{x}_s[\ell]|\mathbf{x}_{s+1}[\ell],\mathbf{x}_0[\ell])\mathrm{d}\mathbf{x}_{s+1}\,.
\end{aligned}
$$

Finally, if $s = \tau_{k+1} - 1$, then

$$
\begin{aligned}
\gamma^{\mathbf{y}}_{0,s}(\mathbf{x}_0,\mathbf{x}_s) &= \mathfrak{p}_0(\mathbf{x}_0)\underline{q}_{s|0}(\underline{x}_s|\underline{x}_0) \prod_{i=1}^{k} q^i_{\tau_i|0}(\mathbf{y}[i]|\mathbf{x}_0[i])q^i_{s|\tau_i}(\mathbf{x}_s[i]|\mathbf{y}[i])\\
&\qquad\times q^{k+1}_{s|0}(\mathbf{x}_s[k+1]|\mathbf{x}_0[k+1])q^{k+1}_{\tau_{k+1}|s}(\mathbf{y}[k+1]|\mathbf{x}_s[k+1]) \prod_{\ell=k+2}^{\mathbf{d_y}} q^\ell_{s|0}(\mathbf{x}_s[\ell]|\mathbf{x}_0[\ell])q^\ell_{\tau_\ell|s}(\mathbf{y}[\ell]|\mathbf{x}_s[\ell])\,,
\end{aligned}
$$

and using

$$
\begin{aligned}
q^{k+1}_{s|0}(\mathbf{x}_s[k+1]|\mathbf{x}_0[k+1])&q^{k+1}_{\tau_{k+1}|s}(\mathbf{y}[k+1]|\mathbf{x}_s[k+1])\\
&= q^{\sigma,k+1}_{s|\tau_{k+1},0}(\mathbf{x}_s[k+1]|\mathbf{x}_{\tau_{k+1}}[k+1],\mathbf{x}_0[k+1])q^{k+1}_{\tau_{k+1}|0}(\mathbf{y}[k+1]|\mathbf{x}_0[k+1])
\end{aligned}
$$

we find that

$$\gamma_{0,s}(\mathbf{x}_0, \mathbf{x}_s)$$

$$= \int \gamma_{0,\tau_{k+1}}^{\mathbf{y}}(\mathbf{x}_0, \mathbf{x}_{\tau_{k+1}}) \underline{q}_{s|\tau_{k+1},0}^{\sigma}(\underline{x}_s|\underline{x}_{\tau_{k+1}}, \underline{x}_0) g_s^{\mathbf{y}}(\overline{x}_s) \prod_{\ell=k+1}^{\mathsf{d_y}} q_{s|s+1,0}^{\sigma,\ell}(\mathbf{x}_s[\ell]|\mathbf{x}_{\tau_{k+1}}[\ell], \mathbf{x}_0[\ell]) \mathrm{d}\mathbf{x}_{\tau_{k+1}} .$$

$\square$

### B.3 ALGORITHMIC DETAILS AND NUMERICS

#### B.3.1 GMM

For a given dimension $\mathsf{d}_x$, we consider $\mathsf{q}_{\mathrm{data}}$ a mixture of 25 Gaussian random variables. The Gaussian random variables have mean $\boldsymbol{\mu}_{i,j} := (8i, 8j, \cdots, 8i, 8j) \in \mathbb{R}^{\mathsf{d}_x}$ for $(i,j) \in \{-2, -1, 0, 1, 2\}^2$ and unit variance. The mixture (unnormalized) weights $\omega_{i,j}$ are independently drawn according to a $\chi^2$ distribution. The $\kappa$ paramater of `MCGdiff` is $\kappa^2 = 10^{-4}$. We use 20 steps of `DDIM` for the numerical examples and for all algorithms.

**Score:** Note that $\mathsf{q}_s(x_s) = \int q_{s|0}(x_s|x_0)\mathsf{q}_{\mathrm{data}}(x_0)\mathrm{d}x_0$. As $\mathsf{q}_{\mathrm{data}}$ is a mixture of Gaussians, $\mathsf{q}_s(x_s)$ is also a mixture of Gaussians with means $\alpha_s^{1/2}\boldsymbol{\mu}_{i,j}$ and unitary variances. Therefore, using automatic differentiation libraries, we can calculate $\nabla \log \mathsf{q}_s(x_s)$. Setting $\mathbf{e}(x_s, s) = -(1 - \alpha_s)^{1/2}\nabla \log \mathsf{q}_s(x_s)$ leads to the optimum of equation 1.4.

**Forward process scaling:** We chose the sequence of $\{\beta_s\}_{s=1}^{1000}$ as a linearly decreasing sequence between $\beta_1 = 0.2$ and $\beta_{1000} = 10^{-4}$.

**Measurement model:** For a pair of dimensions $(\mathsf{d}_x, \mathsf{d_y})$ the measurement model $(y, \mathrm{A}, \sigma_y)$ is drawn as follows:

- A: We first draw $\tilde{\mathrm{A}} \sim \mathcal{N}(0_{\mathsf{d_y} \times \mathsf{d}_x}, \mathrm{I}_{\mathsf{d_y} \times \mathsf{d}_x})$ and compute the SVD decomposition of $\tilde{A} = \mathrm{USV}^T$. Then, we sample for $(i,j) \in \{-2, -1, 0, 1, 2\}^2$, $s_{i,j}$ according to a uniform in $[0, 1]$. Finally, we set $\mathrm{A} = \mathrm{U}\,\mathrm{Diag}(\{s_{i,j}\}_{(i,j)\in\{-2,-1,0,1,2\}^2})\mathrm{V}^T$.

- $\sigma_y$: We draw $\sigma_y$ uniformly in the interval $[0, \max(s_1, \cdots, s_{\mathsf{d_y}})]$.

- $y$: We then draw $x_* \sim \mathsf{q}_{\mathrm{data}}$ and set $y := \mathrm{A}x_* + \sigma_y\epsilon$ where $\epsilon \sim \mathcal{N}(0_{\mathsf{d_y}}, \mathrm{I}_{\mathsf{d_y}})$.

**Posterior:** Once we have drawn both $\mathsf{q}_{\mathrm{data}}$ and $(y, \mathrm{A}, \sigma_y)$, the posterior can be exactly calculated using Bayes formula and gives a mixture of Gaussians with mixture components $c_{i,j}$ and associated weights $\tilde{\omega}_{i,j}$

$$c_{i,j} := \mathcal{N}(\Sigma\left(\mathrm{A}^T y/\sigma_y^2 + \boldsymbol{\mu}_{i,j}\right), \Sigma),$$

$$\tilde{\omega}_i := \omega_i \mathcal{N}(y; \mathrm{A}\boldsymbol{\mu}_{i,j}, \sigma^2 \mathrm{I}_{\mathsf{d}_x} + \mathrm{AA}^T),$$

where $\Sigma := \left(\mathrm{I}_{\mathsf{d}_x} + \sigma_y^{-2}\mathrm{A}^T\mathrm{A}\right)^{-1}$.

**Variational Inference:** The `RNVP` entries in the numerical examination are obtained by Variational Inference using the `RNVP` architecture for the normalizing flow from Dinh et al. (2017). Given a normalizing flow $\mathsf{f}_\phi$ with $\phi \in \mathbb{R}^j, j \in \mathbb{N}_*$, the training procedure consists of optimizing the ELBO, i.e., solving the optimization problem

$$\phi_* = \arg\max_{\phi \in \mathbb{R}^j} \sum_{k=1}^{N_{nf}} \log |\mathrm{Jf}_\phi(\epsilon_i)| + \log \pi_*(\mathsf{f}_\phi(\epsilon_i)), \tag{B.22}$$

where $N_{nf} \in \mathbb{N}_*$ is the minibatch-size, $\mathrm{Jf}_\phi$ the Jacobian of $\mathsf{f}_\phi$ w.r.t $\phi$, and $\epsilon_{1:N_{nf}} \sim \mathcal{N}(0, \mathbf{I})^{\otimes N_{nf}}$. All the experiments were performed using a 10 layers `RNVP`. Equation (B.22) is solved using Adam algorithm Kingma &

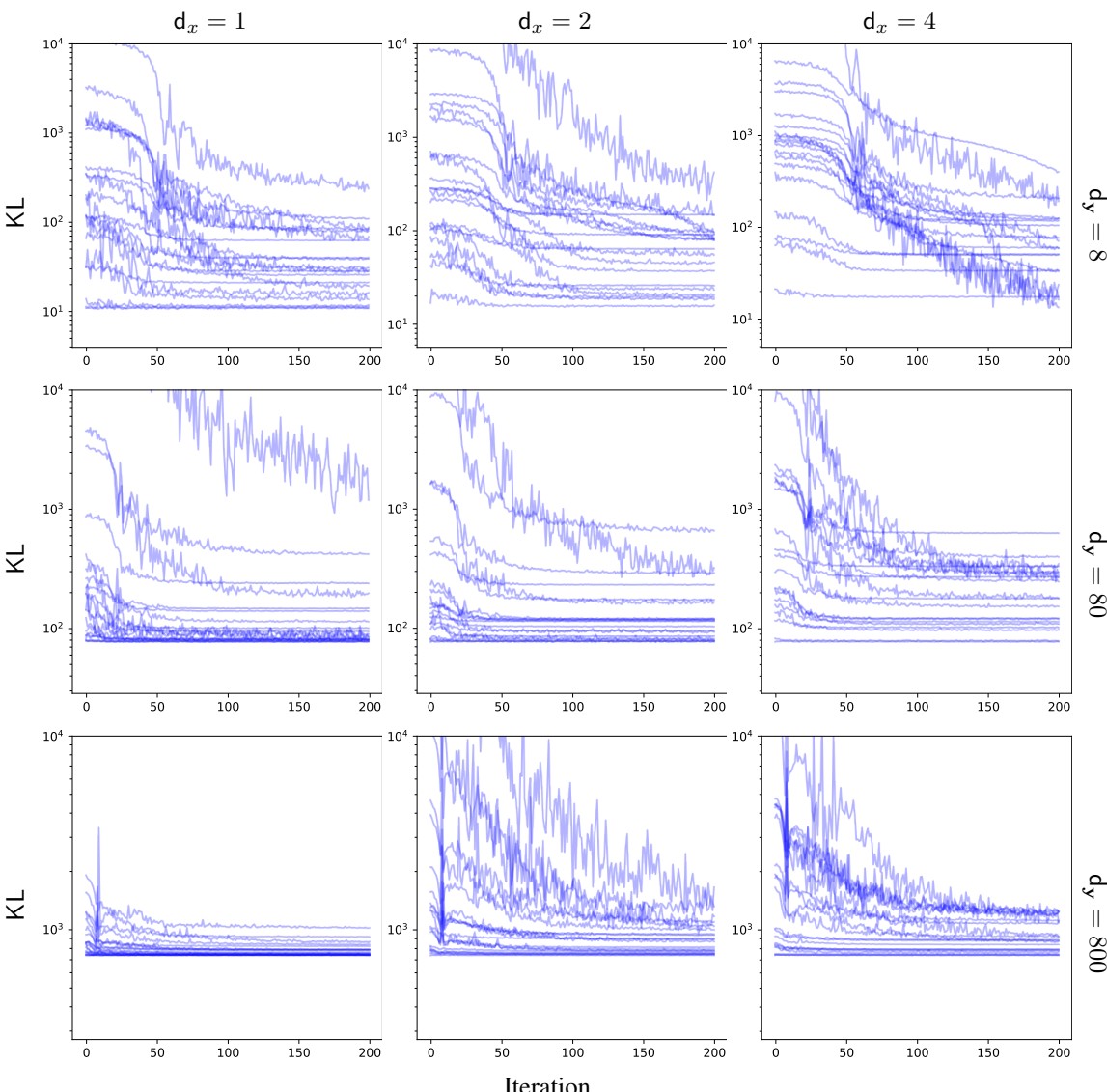

Figure 4: Evolution of KL with the number of iterations for all pairs of $(\mathsf{d}_x, \mathsf{d}_\mathbf{y})$ tested in the GMM case.

Ba (2015) with a learning rate of $10^{-3}$ and 200 iterations with $N_{nf} = 10$. The losses for each pair $(\mathsf{d}_x, \mathsf{d}_\mathbf{y})$ is shown in fig. 4, where one can see that the majority of the losses have converged.

**Choosing `DDIM` timesteps for a given measurement model:** Given a number of `DDIM` samples $R$, we choose the timesteps $1 = t_1 < \cdots < t_R = 1000 \in [1 : 1000]$ as to try to satisfy the two following constraints:

- For all $i \in [1 : \mathsf{d}_\mathbf{y}]$ there exists a $t_j$ such that $\sigma_y \alpha_{t_j}^{1/2} \approx (1 - \alpha_{t_j})^{1/2} s_i$,

- For all $i \in [1 : R - 1]$, $\alpha_{t_i}^{1/2} - \alpha_{t_{i+1}}^{1/2} \approx \delta$ for some $\delta > 0$.

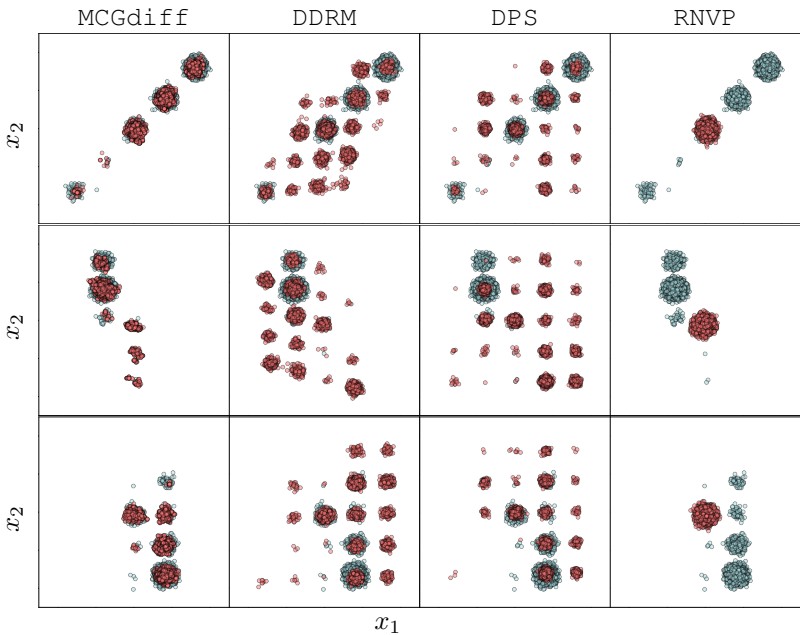

Figure 5: First two dimensions for the GMM case with $d_x = 8$. The rows represent $d_y = 1, 2, 4$ respectively. The blue dots represent samples from the exact posterior, while the red dots correspond to samples generated by each of the algorithms used (the names of the algorithms are given at the top of each column).

The first constraint comes naturally from the definition of $\tau_i$. Since the potentials have mean $\alpha_{t_i}^{1/2} y$, the second condition constrains the intermediate laws remain "close". An algorithm that approximately satisfies both constraints is given below.

---

**Algorithm 2:** Timesteps choice

---

**Input:** Number of DDIM steps $R$, $\sigma_y$, $\{s_i\}_{i=1}^{d_y}$, $\{\alpha_i\}_{i=1}^{1000}$
**Output:** $\{t_j\}_{j=1}^{R}$
Set $S_\tau = \{\}$.
**for** $j \leftarrow [1 : d_y]$ **do**
    Set $\tilde{\tau}_j = \arg\min_{\ell \in [1:1000]} |\sigma_y \alpha_\ell^{1/2} - (1 - \alpha_\ell)^{1/2}) s_j|$.
    Add $\tilde{\tau}_j$ to $S_\tau$ if $\tilde{\tau}_j \notin S_\tau$.
Set $n_m = R - \#S_\tau - 1$ and $\delta = (\alpha_1^{1/2} - \alpha_{1000}^{1/2})/n_m$.
Set $t_1 = 1$, $e = 1$ and $i_e = 1$. **for** $\ell \leftarrow [2 : 1000]$ **do**
    **if** $\alpha_e^{1/2} - \alpha_\ell^{1/2} > \delta$ *or* $\ell \in S_\tau$ **then**
        Set $e = \ell$, $i_e = i_e + 1$ and $\tau_{i_e} = \ell$.
Set $\tau_R = 1000$.

---

**Additional numerics:** We now proceed to illustrate in Figures 5 to 7 the first 2 components for one of the measurement models for all the different combinations of $(d_x, d_y)$ combinations used in table 1. We also show in fig. 8 the evolution of each observed coordinate in the noise case with $d_y = 4$. We can see that it follows closely the forward path of the diffused observations indicated by the blue line.

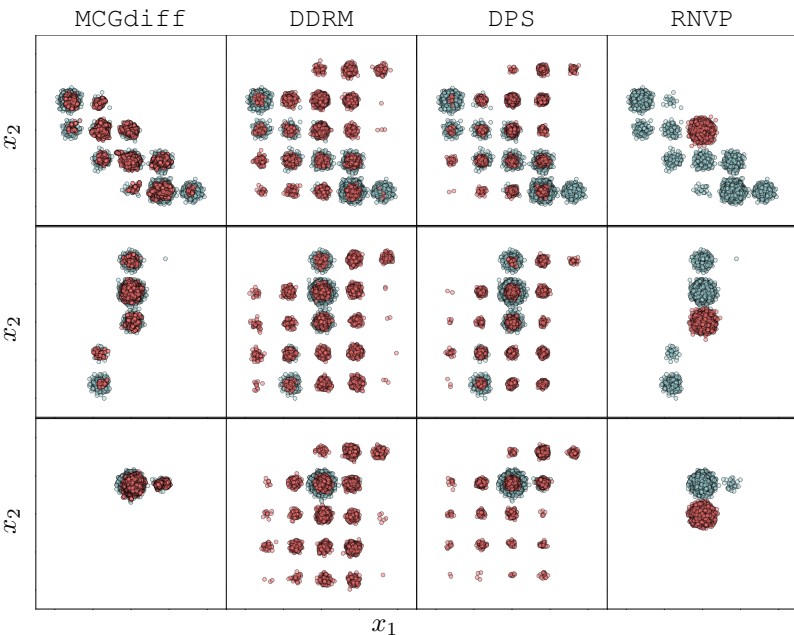

Figure 6: First two dimensions for the GMM case with $d_x = 80$. The rows represent $d_{\mathbf{y}} = 1, 2, 4$ respectively. The blue dots represent samples from the exact posterior, while the red dots correspond to samples generated by each of the algorithms used (the names of the algorithms are given at the top of each column).

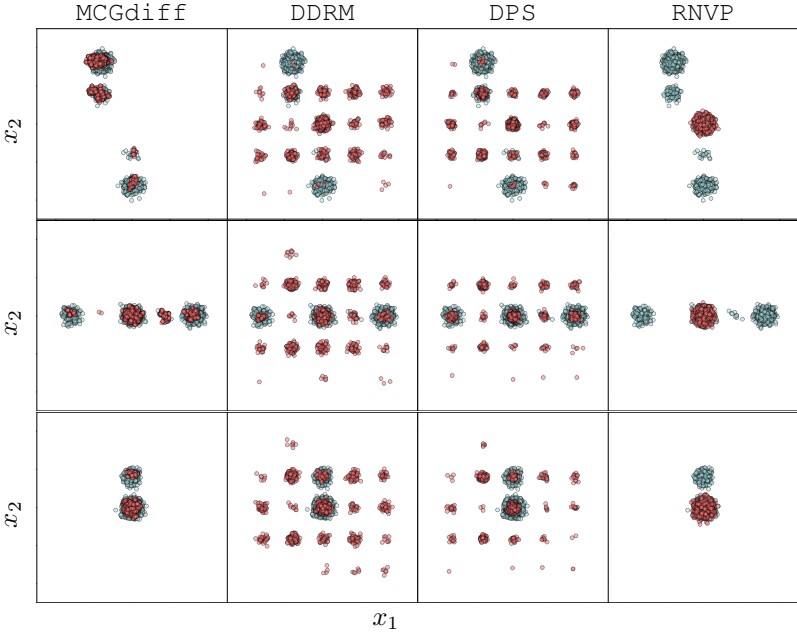

Figure 7: First two dimensions for the GMM case with $d_x = 800$. The rows represent $d_{\mathbf{y}} = 1, 2, 4$ respectively. The blue dots represent samples from the exact posterior, while the red dots correspond to samples generated by each of the algorithms used (the names of the algorithms are given at the top of each column).

| $d$ | $d_y$ | MCGdiff | DDRM | DPS | RNVP |
|-----|-------|---------|------|-----|------|
| 8 | 1 | **1.43 ± 0.55** | 5.88 ± 1.16 | 4.86 ± 1.01 | 9.43 ± 0.99 |
| 8 | 2 | **0.49 ± 0.24** | 5.20 ± 1.32 | 5.79 ± 1.96 | 8.93 ± 1.29 |
| 8 | 4 | **0.38 ± 0.25** | 2.51 ± 1.29 | 3.48 ± 1.52 | 6.71 ± 1.54 |
| 80 | 1 | **1.39 ± 0.45** | 5.64 ± 1.10 | 4.98 ± 1.14 | 6.86 ± 0.88 |
| 80 | 2 | **0.67 ± 0.24** | 7.07 ± 1.35 | 5.10 ± 1.23 | 7.79 ± 1.50 |
| 80 | 4 | **0.28 ± 0.14** | 7.81 ± 1.48 | 4.28 ± 1.26 | 7.95 ± 1.61 |
| 800 | 1 | **2.40 ± 1.00** | 7.44 ± 1.15 | 6.49 ± 1.16 | 7.74 ± 1.34 |
| 800 | 2 | **1.31 ± 0.60** | 8.95 ± 1.12 | 6.88 ± 1.01 | 8.75 ± 1.02 |
| 800 | 4 | **0.47 ± 0.19** | 8.39 ± 1.48 | 5.51 ± 1.18 | 7.81 ± 1.63 |

Table 2: Extended GMM sliced wasserstein table.

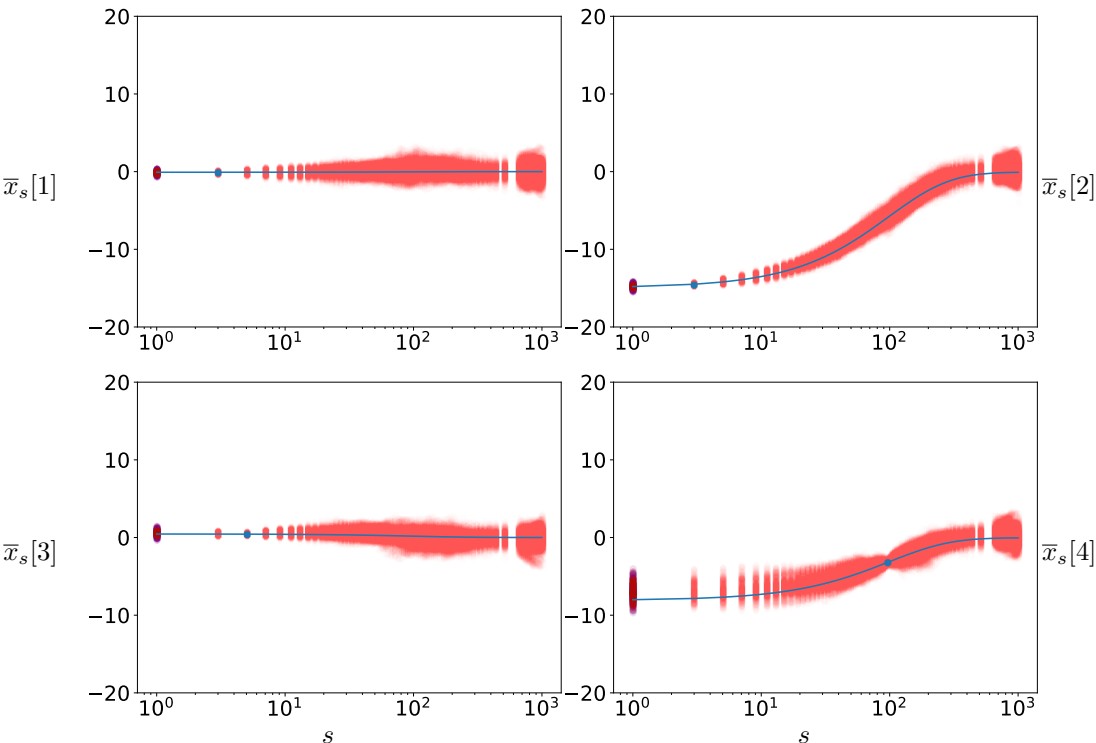

Figure 8: Illustration of the particle cloud of the 4 first observed coordinate in the case $(\mathsf{d_y}, \mathsf{d}_x) = (4, 800)$ with 100 DDIM steps. The red points represent the particle cloud, while the purple points at the origin represent the posterior distribution. The blue curve corresponds to the curve $s \to \alpha_s^{1/2} \mathbf{y}[\ell]$ and the blue dot on the curve to $\alpha_{\tau_\ell}^{1/2} \mathbf{y}[\ell]$.

Table 2 is an extended version of table 1.

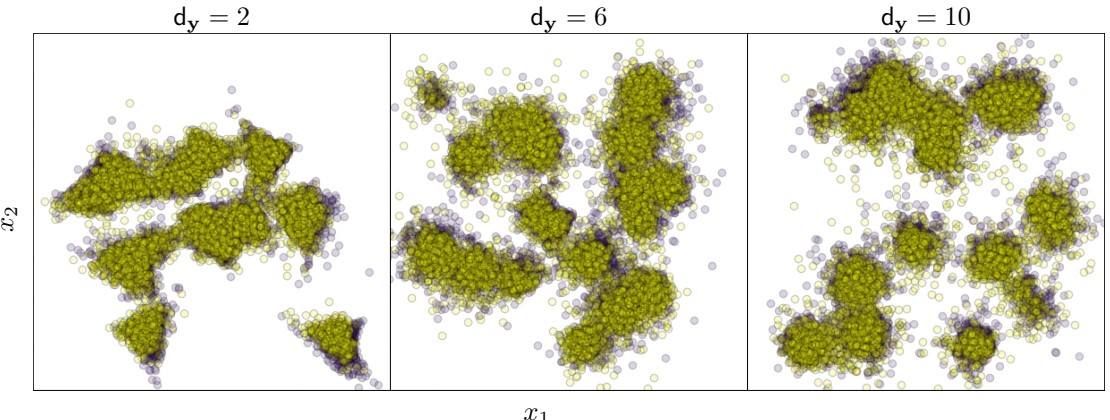

Figure 9: Purple points are samples from the prior and yellow samples from the diffusion with $25$ `DDIM` steps.

| $d$ | SW |
|---|---|
| 2 | $0.79 \pm 0.15$ |
| 6 | $0.87 \pm 0.07$ |
| 10 | $0.96 \pm 0.06$ |

Table 3: Sliced Wasserstein between learned diffusion and target prior.

### B.3.2 FMM

A funnel distribution is defined by the following density

$$\mathcal{N}(x_1; 0, 1) \prod_{i=1}^{d} \mathcal{N}(x_i; 0, \exp(x_1/2)) .$$

To generate a Funnel mixture model of 20 components in dimension $d$, we start by firstly sampling $(\mu_i, R_i)_{i=1}^{20}$ uniformly in $([-20, 20]^d \times \mathsf{SO}(R^d))^{\times 20}$. The mixture will consist of 20 Funnel random variables translated by $\mu_i$ and rotated by $R_i$, with unnormalized weights $\omega_{i,j}$ that are independently drawn uniformly in $[0, 1]$.

**Score** The denoising diffusion network $\mathbf{e}(\theta)$ in dimension $d$ is defined as a 5 layers Resnet network where each Resnet block consists of the chaining of three blocks where each block has the following layers:

- Linear $(512, 1024)$,
- 1d Batch Norm,
- ReLU activation.

The Resnet is preceeded by an input embedding from dimension $d$ to $512$ and in the end an output embedding layer projects the output of the resnet from $512$ to $d$. The time $t$ is embedded using positional embedding into dimension $512$ and is added to the input at each Resnet block. The network is trained using the same loss as in Ho et al. (2020) for $10^4$ iterations using a batch size of $512$ samples. A learning rate of $10^{-3}$ is used for the Adam optimizer Kingma & Ba (2015). Figure 9 illustrate the outcome of the learned diffusion generative model and the target prior. In table 3 we show the CLT $95\%$ intervals for the SW between the learned diffusion generative model and the target prior.

**Forward process scaling** We chose the sequence of $\{\beta_s\}_{s=1}^{1000}$ as a linearly decreasing sequence between $\beta_1 = 0.2$ and $\beta_{1000} = 10^{-4}$.

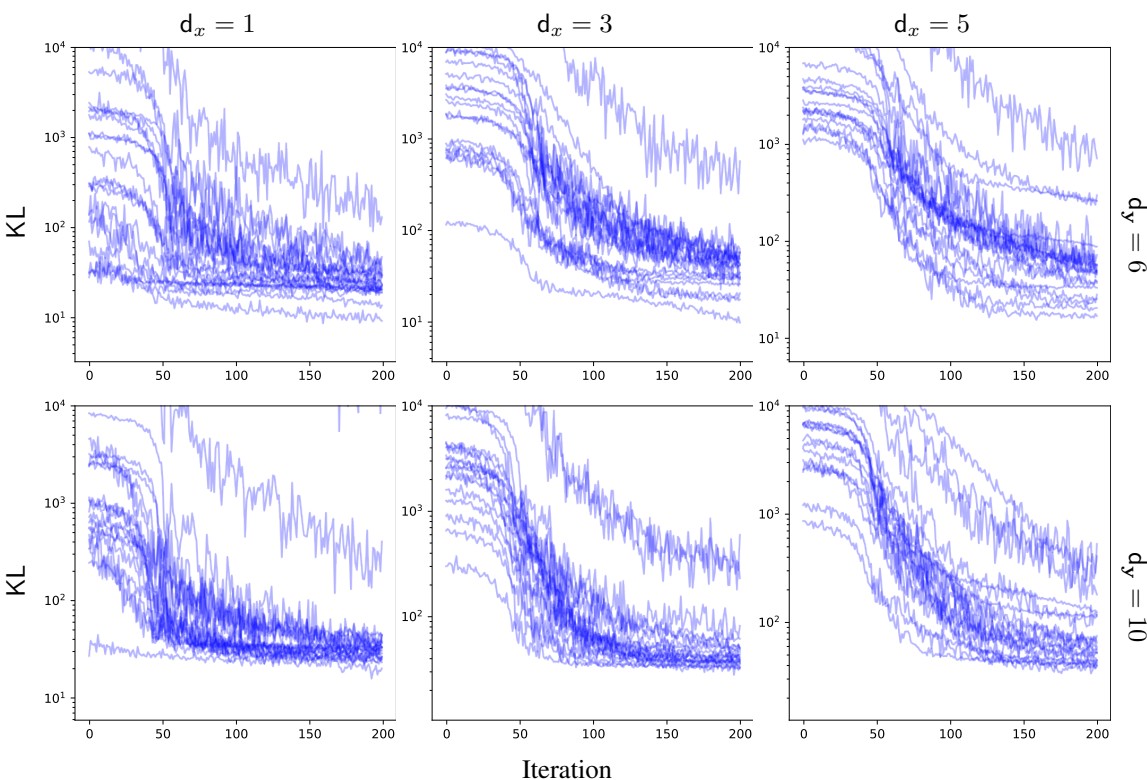

Figure 10: Evolution of KL with the number of iterations for all pairs of $(\mathsf{d}_x, \mathsf{d_y})$ tested in the FMM case.

**Measurement model**    The measurement model was generated in the same way as for the GMM case.

**Posterior**    The posterior samples were generated by running the No U-turn sampler (Hoffman & Gelman (2011)) with a chain of length $10^4$ and taking the last sample of the chain. This was done in parallel to generate $10^4$ samples. The mass matrix and learning rate were set by first running Stan's warmup and taking the last values of the warmup phase.

**Variational inference:**    Variational inference in FMM shares the same details as the GMM case. The analogous of fig. 4 is displayed at fig. 10.

**Additional plots:**    We now proceed to illustrate in Figures 11 to 13 the first 2 components for one of the measurement models for all the different combinations of $(\mathsf{d}_x, \mathsf{d_y})$ combinations used in table 1.

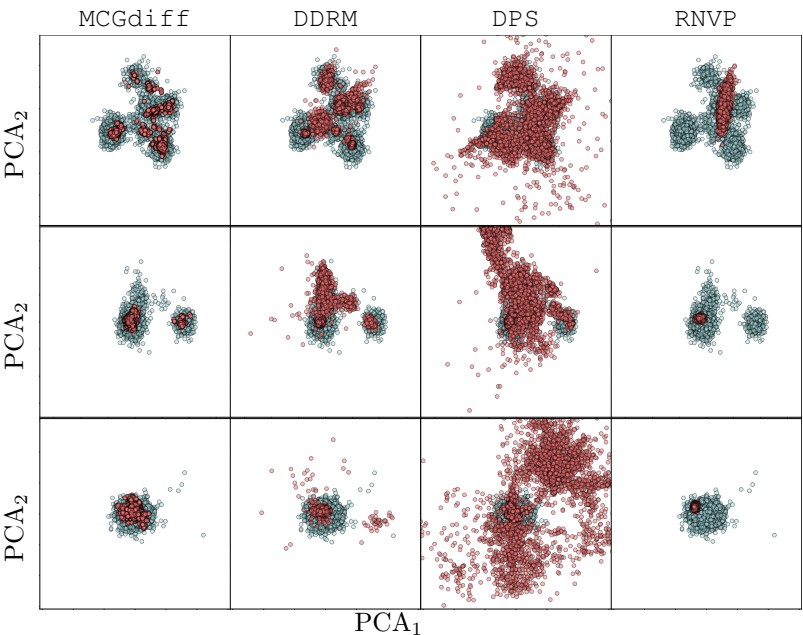

Figure 11: First two dimensions for the FMM case with $d_x = 10$. The rows represent $d_y = 1, 3, 5$ respectively. The blue dots represent samples from the exact posterior, while the red dots correspond to samples generated by each of the algorithms used (the names of the algorithms are given at the top of each column).

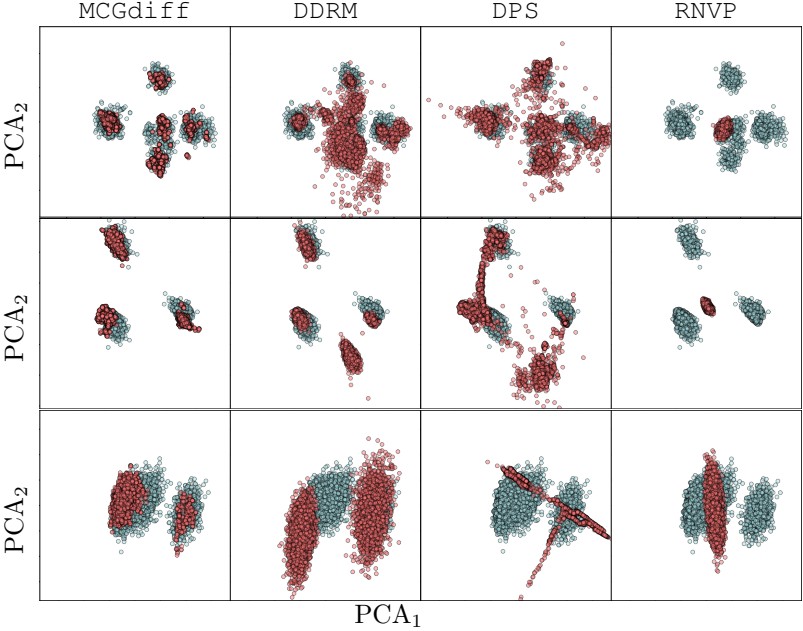

Figure 12: First two dimensions for the FMM case with $d_x = 6$. The rows represent $d_y = 1, 3, 5$ respectively. The blue dots represent samples from the exact posterior, while the red dots correspond to samples generated by each of the algorithms used (the names of the algorithms are given at the top of each column).

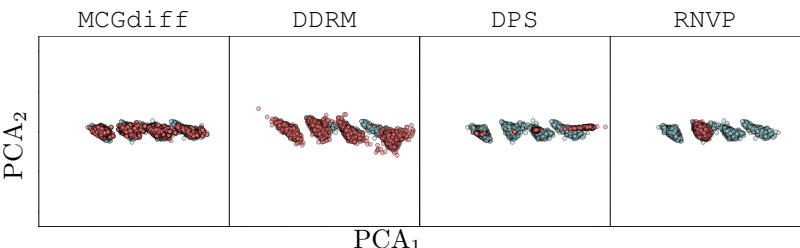

Figure 13: First two dimensions for the FMM case with $d_x = 2$ and $d_\mathbf{y} = 1$. The blue dots represent samples from the exact posterior, while the red dots correspond to samples generated by each of the algorithms used (the names of the algorithms are given at the top of each column).

### B.3.3 IMAGE DATASETS

We now present samples from MCGdiff in different image dataset and different kinds of inverse problems.

**Super Resolution** We start by super resolution. We set $\sigma_y = 0.05$ for all the datasets and $\zeta_{\text{coeff}} = 0.1$ for DPS . We use 100 steps of DDIM with $\eta = 1$. The results are shown in Figure 14. We use a downsampling ratio of 4 for the CIFAR-10 dataset, 8 for both Flowers and Cats datasets and 16 for the others. The dimension of the datasets are recalled in table 4. We display in fig. 14 samples from MCGdiff, DPS and DDRMover several different image datasets (table 4). For each algorithm, we generate 1000 samples and we show the pair of samples that are the furthest apart in $L^2$ norm from each other in the pool of samples. For MCGdiff we ran several parallel particle filters with $N = 64$ to generate 1000 samples.

|  | CIFAR-10 | Flowers | Cats | Bedroom | Church | CelebaHQ |
|---|---|---|---|---|---|---|
| $(W, H, C)$ | $(32, 32, 3)$ | $(64, 64, 3)$ | $(128, 128, 3)$ | $(256, 256, 3)$ | $(256, 256, 3)$ | $(256, 256, 3)$ |

Table 4: The datasets used for the inverse problems over image datasets.

**Gaussian 2D debluring** We consider a Gaussian 2D square kernel with sizes $(w/6, h/6)$ and standard deviation $w/30$ where $(w, h)$ are the width and height of the image. We set $\sigma_y = 0.1$ for all the datasets and $\zeta_{\text{coeff}} = 0.1$ for DPS . We use 100 steps of DDIM with $\eta = 1$. We display in fig. 15 samples from MCGdiff, DPS and DDRMover several different image datasets (table 4). For each algorithm, we generate 1000 samples and we show the pair of samples that are the furthest appart in $L^2$ norm from each other in the pool of samples. For MCGdiff we ran several parallel particle filters with $N = 64$ to generate 1000 samples.

**Inpainting on CelebA** We consider the inpainting problem on the CelebA dataset with several different masks in fig. 16. We show in fig. 17 the evolution of the particle cloud with $s$.

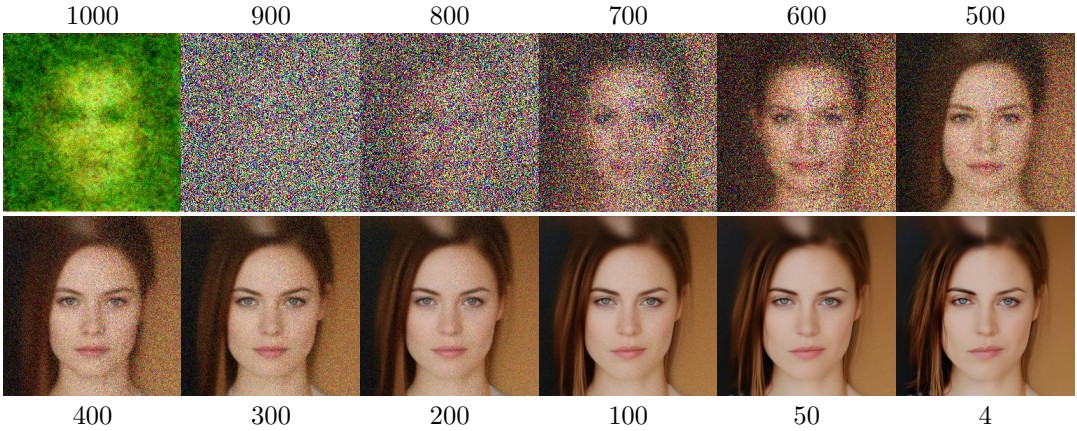

Figure 17: Evolution of the particle cloud for one of the masks. The numbers on top and bottom indicate the step $s$ of the approximation.

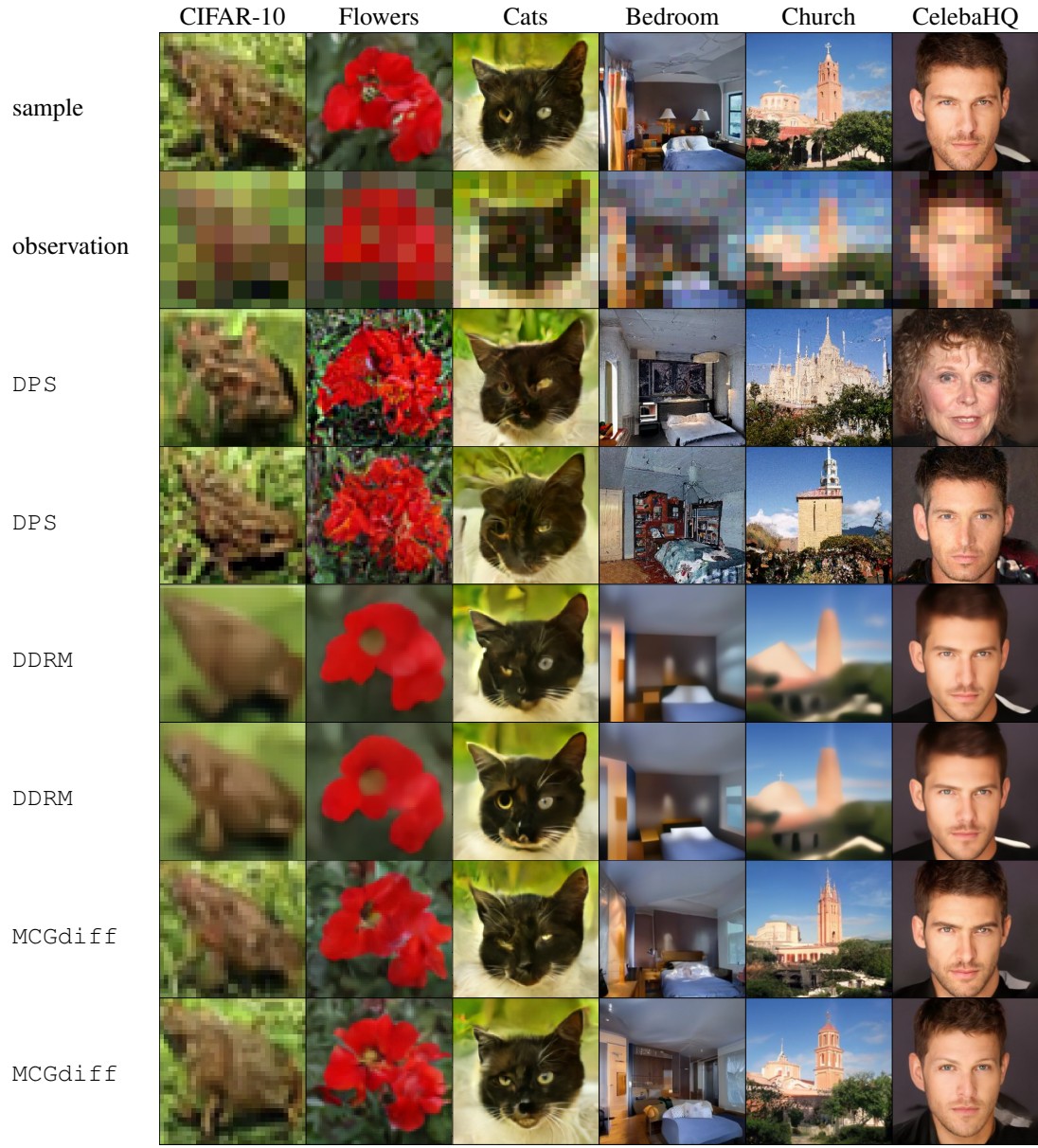

Figure 14: Ratio 4 for CIFAR, 8 for flowers and Cats and 16 for CELEB

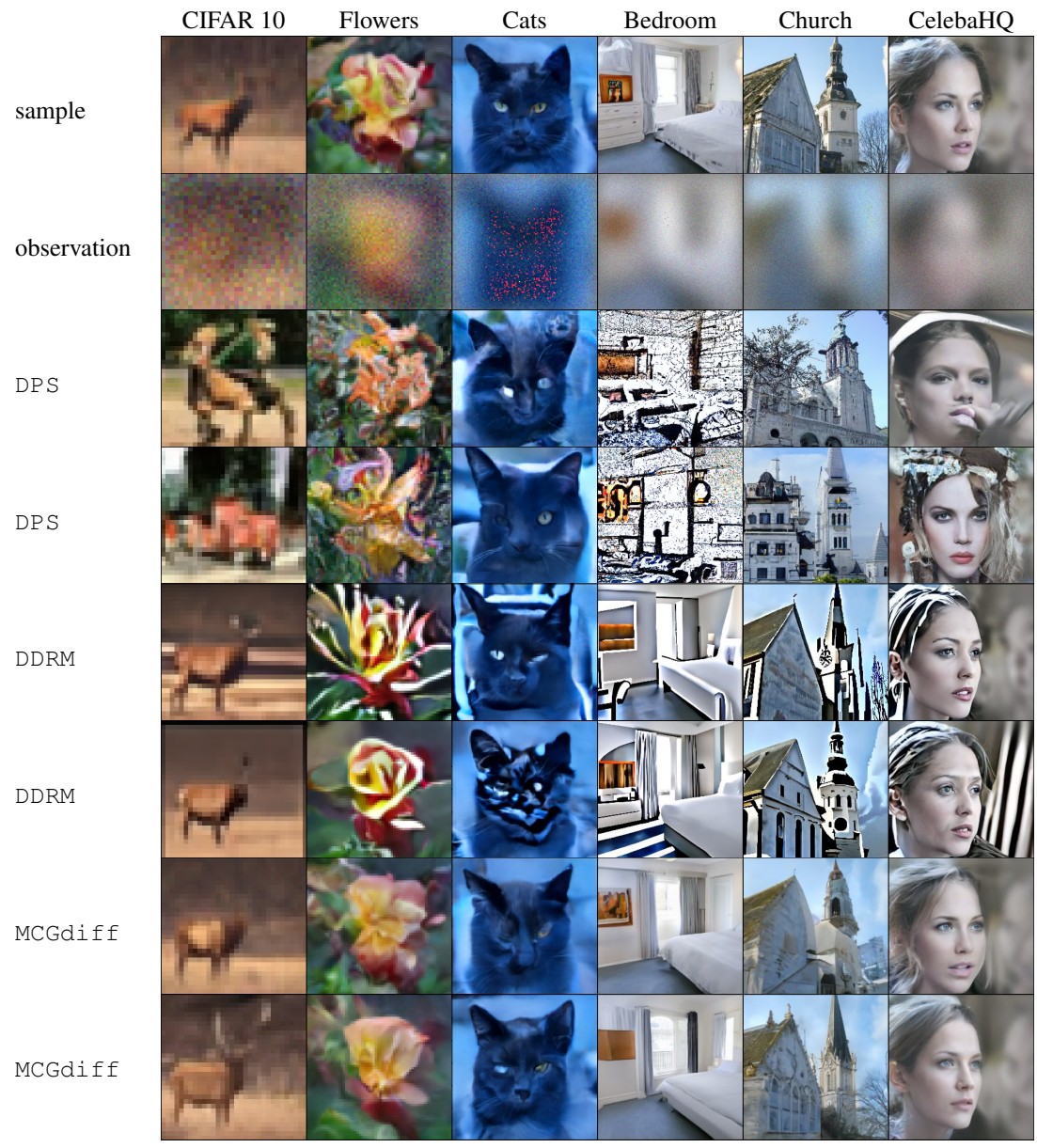

Figure 15

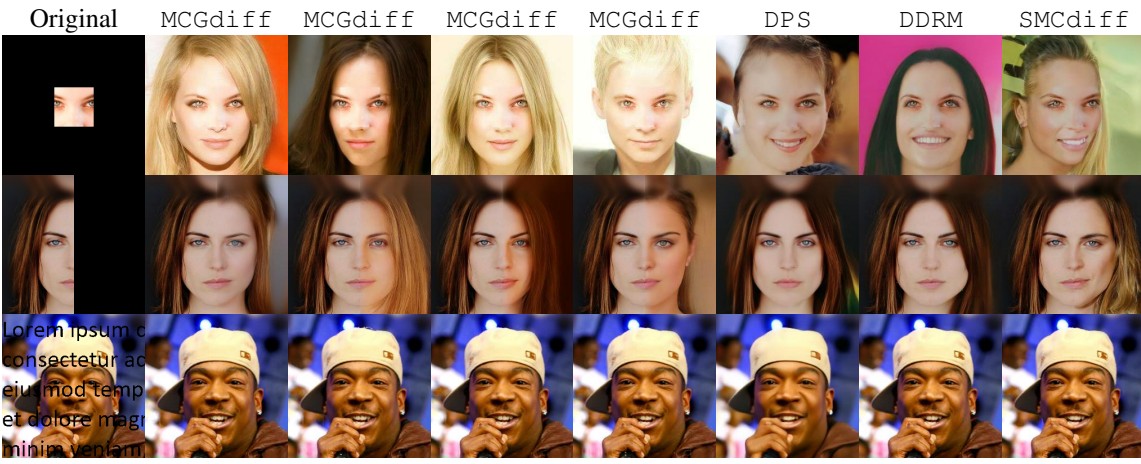

Figure 16: Inpainting with different masks on the CelebA test set.

