# OpenReview forum: "Monte Carlo guided Denoising Diffusion models for Bayesian linear inverse problems."
_ICLR.cc/2024/Conference — ICLR 2024 oral_

### Official Review · Reviewer_68J7 · 2023-10-30

**Soundness:** 3 good
**Presentation:** 3 good
**Contribution:** 3 good
**Rating:** 6
**Confidence:** 4

**Summary:**

This work proposed an interesting method that combines particle filtering with diffusion model, trying to mitigate the issue of ill-posed inverse problem. The idea is novel and the demonstration of the proposed algorithm looks good in two dimensional illustrations, but the results in image datasets doesn't give clear distinctions from other existing methods.

**Strengths:**

1. Interesting novel idea to integrate particle filtering method into the exploration of posterior under the diffusion model framework.

2. Theoretical work looks solid and convincing.

**Weaknesses:**

1. It seems the contribution point 3 is mainly a support for the first two points, hardly it can be classified as an independent contribution.

2. A pseudo-code or diagram describing the idea of this method would make it much easier to be interpreted

3. This method should be pretty slow as both diffusion model and particle filter are computationally heavy ones. Would be good to clearly state the limitation and also report the runtime etc.

**Questions:**

1. It seems very difficult to conclude that the proposed method outperforms the others in image dataset, is there any other ways to quantitatively  demonstrate the advantage of the proposed method?

2. Does the intrinsic degeneracy issue from particle filter affect the stability and performance of the proposed method?

---

> ### Author Response · Authors · 2023-11-19
>
> Dear reviewer,
> We thank you for the time and effort taken to review our manuscript. Regarding your comments:
> - We think that the point 3 is an empirical contribution in itself since we show that the current methods used for solving linear inverse problems with diffusion do not actually correctly sample the posterior in the most straightforward cases.
> - While we agree that our method is computationally more expensive than existing works such as DDRM and DPS, we believe that this is the price to pay to obtain more faithful approximations of the posterior. Indeed, as we show in the Gaussian mixture experiment, DPS and DDRM do not actually sample approximately from the posterior and cannot be made more precise using more computational resources. On the other hand, our method has an extra parameter (the number of particles $N$) which controls the accuracy of the approximation. We firmly believe that there is an intrinsic tradeoff between accuracy and computational burden when dealing with the posteriors of generative models. That being said, in moderate dimensions, drawing $N$ parallel samples from DPS or DDRM has the same computational cost as running one particle filter with $N$ particles, which returns $N$ samples that approximate the posterior (the resampling cost is negligible in front of the cost of running the diffusion). In much larger dimensions, taking $N$ particles in MCGDiff will result in $N$ collapsed particles, which is not qualitatively the same thing as running DPS or DDRM with $N$ samples. However, we advocate in our experiments that in such cases, one should use a small number of particles and instead run $M$ parallel MCGDiff with $N/M$ particles, to obtain a particle approximation of the posterior with $M$ different and independent particles. In such cases, the computational cost of our algorithm is larger than that of DPS and DDRM (in the sense that with the same computational cost you can draw $N$ i.i.d. samples from DPS and DDRM). While it is true that the particles from a single particle filter lack diversity in global level features, still, we believe that the quality of our proposal allows us to draw samples in the support of the posterior, and not outside, as it is the case of the other methods. Hence running parallel particle filters allows us to have samples that are globally diverse.
> - For our imaging experiments, we believe that no method can quantitatively assert the performances of all three algorithms, as we do not have samples from the posterior to compare to. This is precisely the reason why we focus on tractable experiments in the main paper. In the appendix though, we have compared MCGDiff and other algorithms in some image tasks by at first generating 100 samples from the posterior and then showing the 2 samples from those that are the furthest apart. We can see that MCGDiff seems to obtain more coherent samples, even though, as we said before, this is purely a qualitative evaluation and is of course subject to discussion.

---

> ### Comment · Reviewer_68J7 · 2023-11-22
>
> Thanks for the response, that clarifies the computational question I posted, although I am still not fully convinced by the claim that 'extra parameter (the number of particles N) which controls the accuracy of the approximation', increasing the number of particles will certainly increase computational cost but not necessarily improving the accuracy, my concern is still on how the intrinsic degeneracy issue in particle filter will affect the whole algorithm. I would suggest adding 'limitations of the method' to your final version. Although I still have minor concern as described above, the novelty and the soundness of this work is good, I will keep my weak accept score.

---

### Official Review · Reviewer_3cyr · 2023-10-31

**Soundness:** 4 excellent
**Presentation:** 4 excellent
**Contribution:** 4 excellent
**Rating:** 10
**Confidence:** 3

**Summary:**

A method is proposed to solve linear inverse problems using score-based generative models (SGM), i.e. denoising diffusion models, in a Bayesian manner using Sequential Monte Carlo (SMC).  Such statistical approaches allow one to sample from the posterior, or an approximation of it, which facilitates uncertainty quantificaiton.  Experiments are presented on simple distributions where the posterior is known, where validation can be performed.  Further experiments are then presented on real images to demonstrate practical application, although underlying true posteriors for validation are not available in these settings.  Anonymized code is made available.

**Strengths:**

Solving inverse problems using SGMs is a topical area of research at present, with a number of recent papers.  The key problem is to faithfully sample from the underlying posterior distribution.  It is not straightforward to integrate a likelihood into SGMs since it is difficult to consider a closed-form for the likelihood due to the dependence on time and thus noise level.  Existing approaches address this issue by various approximations (e.g. data consistency projections) but as far as I'm aware, and as the authors also state, no existing method fully solves this problem.  The authors claim they present the first provably consistent algorithm for condional sampling from the target posterior.  While I haven't checked all of the mathematical details I believe this is indeed the case.  Numerical experiments indeed confirm for the simple distributions that the sliced Wasserstein distance between the proposed method and the posterior recovered either analytically or by NUTS sampling is considerably smaller than for other SGM approaches for solving inverse problems.

**Weaknesses:**

The authors comment that standard image metrics, e.g. FID, are not suitable for evaluating Bayesian reconstruction methods for solving inverse problems.  Results from a handful of examples of inverse imaging problems are presented, which all look very compelling.  Nevertheless, it would be useful to summarise performance over a larger set of test images, if possible.

**Questions:**

Could the authors propose a metric to summarise performance over a large set of test images (see discussion in Weaknesses)?

---

> ### Author Response · Authors · 2023-11-21
>
> Dear reviewer,
>
> We would like to start by thanking you for the time and effort you put into reviewing our paper. We are glad that you have appreciated our work!
>
> Indeed, FID is not suitable for Bayesian reconstruction problems and as far as we are concerned, there are no other consistent methods that permit a reliable assessment of such reconstructions. Note that in the particular case where the posterior is concentrated, it is possible to “confidently” compare different reconstruction algorithms, through the LPIPS metric using the groundtruth image. However, we have been particularly interested in ill-posed problems, where this type of comparisons is not relevant. One line of exploration could be to run MCGDiff with a very large number of particles and use the resulting samples as a groundtruth with some metric over distributions, for example the sliced Wasserstein distance.

---

> > ### Comment · Reviewer_3cyr · 2023-11-23
> >
> > Thank you for the additional comments regarding further validation.  The line of exploration that you suggest sounds interesting but is not a requirement for submission.  I will keep my current strong accept rating.

---

### Official Review · Reviewer_3L38 · 2023-11-06

**Soundness:** 4 excellent
**Presentation:** 3 good
**Contribution:** 4 excellent
**Rating:** 8
**Confidence:** 4

**Summary:**

This article propose a sequential Monte-Carlo method to solve linear inverse problems such as deblurring, super-resolution or inpainting with score-based generative priors (aka generative diffusion models) through the design of an efficient sampler.  The method is embodied into their proposed MCGdiff algorithm, which is proved to be sampling conditionally in a consistent manner from the diffusion posteriors. They evaluate the performance of their algorithm on various numerical simulation, demonstrating state of the art results on several imaging inverse problem applications.

**Strengths:**

The paper is dense but accessible and sufficiently well written, and although terse at time, thorough and complete. It is also well illustrated althought Fig. 1 is more puzzling than useful. The theory is well-developed, complete with all the proofs.

In terms of results,  MCGdiff seems to provide very good quality samples in all illustrated problems in comparison with other diffusion-based models. It interesting to see it perform well on difficult cases.

Code is available.

**Weaknesses:**

The paper is detailed and the appendices can be hard to read.
The authors don't really solve inverse problems in the traditional sense. They generate realistic samples learned from a distribution that are consistent with the observations. While this sounds exactly like solving inverse problems, the resulting images, although better than existing methods and very sharp and detailed, only resemble the ground truth. I mention this because these types of methods cannot yet be used in sensitive contexts like medical imaging or science in general.
There is a lack of control and interpretability on the generated images, as with all the current diffusion methods.
The code issues have not been addressed. From looking at the source, the diffusion code seems to be based on DDPM, which is not acknowledged in the main text or appendices. Speed issues have not been mentioned.
The bibliography is excellent except for the first paragraph of the introduction. Linear inverse problems as described have been studied mathematically for a very long time (Fredholm, etc) and in computer vision since the late 1980s at least. Perhaps the bibliography should mention this.

**Questions:**

- How could the results be made more interpretable or controllable.
- The experiments on GM seem to indicate that MCGdiff does sample from he posterior distribution but not necessarily in a "thorough manner" see Fig. 5 and 6 ; how can this be better handled?
- How do deal with noise present in the training dataset ?

---

> ### Author Response · Authors · 2023-11-18
>
> Dear reviewer,
> We thank you for the time and effort taken to review our manuscript and the positive feedback. We thank you for the points that you raised concerning the bibliography and the speed issues and will be integrating it into the main text in the future. We will also add a precise pseudo code that reflects exactly the source code, in the appendix.
>
> - We would like to point out that we refrain from comparing to the ground truth. Indeed, our main focus on the numerical part is indeed the mixture of gaussians and mixture of funnel cases. The main reason is that in those cases it is possible to have access to samples from the real posterior and we show that our algorithm outperforms the competition in all settings and manages to match well the posterior.
>
> - Indeed in figure 5 and 6 we see that unlike the other existing methods, MCGDiff finds all the modes but in some cases the tails of each mode are thinner. A straightforward answer to your question, relying on the theorems presented in the main paper, would be to increase the number of particles being used. There are also other alternatives, as for example using the unadjusted Langevin algorithm initialized at the particles resulting from MCGDiff and targeting the posterior of the model. This requires having the score at time 0, to which we may not have access as the data distribution may not have a density in practice. However, we can replace it with the score at some timestep epsilon, close to 0. We have tried this method and in some cases it leads to better spreading around the mode and smaller sliced Wasserstein distance.
>
> - The current work is an effort to handle bayesian linear inverse problems when using a pre-trained denoising diffusion generative model as a prior, without any need of (further) training. In this sense, the measurement $y$ can be a noisy linear observation of the state. But if we understand your question correctly, it concerns the training data for the underlying generative model, which we do not have control over as we use off the shelf trained diffusion models. However, if one has some prior information on the noise present in the dataset, perhaps there might a path forward. This is a really interesting yet difficult question that we believe is outside the scope of the current paper.

---

### Official Review · Reviewer_3c1h · 2023-11-09

**Soundness:** 4 excellent
**Presentation:** 4 excellent
**Contribution:** 4 excellent
**Rating:** 10
**Confidence:** 2

**Summary:**

The authors introduce a method for addressing ill-posed inverse problems by employing a diffusion-based neural network model guided by Monte Carlo sampling. They refer to their approach as the MCGDiff (Monte Carlo Guided Diffusion) algorithm. Their algorithm is specifically applied to Score-based Generative Models (SGMs) and is used for various image-related tasks, including inpainting, super-resolution, deblurring, and colorization. To model prior distributions, the authors utilize Gaussian Mixed Models (GMM) and Funnel Mixture Models (FMM).

**Strengths:**

A substantial portion of the research paper is devoted to providing a comprehensive background on the use of diffusion models for solving ill-posed inverse problems. Additionally, the paper includes detailed mathematical proofs regarding the algorithm's performance, covering both noiseless and noisy cases. This comprehensive coverage enhances the clarity and rigor of the research.

The authors' consideration of the broader applicability of their work beyond their experimental investigation of image data is commendable. It highlights the potential relevance and impact of their findings in a wider context of inverse problems.

The comparative results presented in the image context are particularly noteworthy. The research demonstrates impressive performance, especially in the way the generated posterior sampling distributions align with the exact ones. This underscores the effectiveness and accuracy of the proposed approach when compared to competing methods.

**Weaknesses:**

It would have been intriguing to explore the performance of the MCGDiff algorithm on non-image data. While the research focuses on image-related tasks, extending the investigation to other data types would provide a broader perspective on the algorithm's applicability and effectiveness across various domains.

**Questions:**

1. Are there any plans to release your code for the MCGDiff algorithm to facilitate further research and practical applications in the broader community?

2. Have you explored the application of the MCGDiff algorithm in addressing inverse problems beyond image-related tasks, and if so, could you share any insights into its performance and adaptability in those contexts?

---

> ### Author Response · Authors · 2023-11-18
>
> Dear reviewer,
> We thank you for the time and effort taken to review our manuscript. We are glad that you have appreciated our work and thank you for the positive feedback!
>
>  The code that is currently available in the anonymous git will be available once the submission process is over. Indeed, we also believe that MCGDiff can have a broader impact beyond image-related tasks. We are currently using it on healthy electrocardiogram (ECG) to develop interpretable outlier detection tools for electrophysiological anomalies, with great success.

---

> > ### Comment · Reviewer_3c1h · 2023-11-22
> >
> > Your current exploration of healthcare applications is truly inspiring, both on a general level and as a topic of personal interest to me. I eagerly anticipate keeping a watchful eye on arXiv for the release of this work. It's exciting to see the potential broader impact of MCGDiff beyond image-related tasks, particularly in the successful development of interpretable outlier detection tools for electrophysiological anomalies in healthy electrocardiograms (ECG). My rating stands.

---

### Meta-Review · Area_Chair_ETf9 · 2023-12-08

**Metareview:**

This paper presents MCGDiff (Monte Carlo Guided Diffusion) for solving ill-posed linear inverse problems. The method integrates Sequential Monte Carlo (SMC) techniques with score-based generative models (SGMs) to address these problems. MCGDiff is designed to efficiently sample from the posteriors of a sequence of intermediate linear inverse problems, each with decreasing noise levels.

The approach uses Gaussian Mixed Models (GMM) and Funnel Mixture Models (FMM) to model prior distributions. Numerical simulations demonstrate the performance of the method in various image-related tasks such as inpainting, super-resolution, deblurring, and colorization. The paper also discusses experiments on simple distributions where the posterior is known, allowing for validation. The authors do provide anonymized code .

**Justification For Why Not Higher Score:**

N/A

**Justification For Why Not Lower Score:**

To the best of my knowledge, the proposed method is the first provably consistent algorithm for conditional sampling with diffusion models.

---

### Decision · Program_Chairs · 2024-01-16

Accept (oral)